# Enhancement of Polyvinyl Alcohol-Based Films by Chemically Modified Lignocellulosic Nanofibers Derived from Bamboo Shoot Shells

**DOI:** 10.3390/polym17111571

**Published:** 2025-06-05

**Authors:** Jingjing Du, Jianlong Guo, Qian Zhu, Jiagang Guo, Jiayu Gu, Yuhan Wu, Ling Ren, Song Yang, Jian Jiang

**Affiliations:** 1Institute of Agro-Products Processing, Anhui Academy of Agricultural Sciences, Hefei 230041, China; dujjmm@126.com (J.D.); zhuqian0816@163.com (Q.Z.); guojiagang@163.com (J.G.); gujiayu2018@126.com (J.G.); wuyuhan63@163.com (Y.W.); 2Anhui Engineering Laboratory for Functional Microorganisms and Fermented Foods, Anhui Academy of Agricultural Sciences, Hefei 230041, China; 3College of Food Science and Engineering, Anhui Science and Technology University, Chuzhou 239000, China; jianlong_guo@163.com; 4Huaibei Shunfa Food Co., Ltd., No. 10 Qianlong Avenue, Huaibei 235100, China; ren-ling111@163.com

**Keywords:** bamboo shoot shell, lignocellulosic nanofiber, PVA, chemical modification, agricultural waste valorization

## Abstract

In this study, polyvinyl alcohol (PVA) films were reinforced with lignocellulosic nanofibers (LCNFs) extracted from bamboo shoot shells using a choline chloride-based deep eutectic solvent (DES). A filler loading of 10 wt% was identified as the optimal condition for enhancing film performance. To improve interfacial compatibility between the PVA matrix and LCNFs, three surface modification treatments were applied to the nanofibers: hydrochloric acid (HCl) hydrolysis, citric acid (CA) crosslinking, and a dual modification combining both methods (HCl&CA). Among all formulations, films incorporating dual-modified LCNF at 10 wt% loading exhibited the most significant improvements. Compared to neat PVA, these composites showed a 79.2% increase in tensile strength, a 15.1% increase in elongation at break, and a 33.1% enhancement in Young’s modulus. Additionally, thermal stability and barrier properties were improved, while water swelling and solubility were reduced. Specifically, the modified films achieved a thermal residue of 9.21% and the lowest degradation rate of 10.81%/min. Water vapor transmission rate and oxygen permeability decreased by 18.8% and 18.6%, respectively, and swelling and solubility dropped to 14.26% and 3.21%. These results highlight the synergistic effect of HCl hydrolysis and CA crosslinking in promoting uniform filler dispersion and strong interfacial adhesion, offering an effective approach to valorizing bamboo shoot shell waste into high-performance, eco-friendly packaging materials.

## 1. Introduction

The growing emphasis on sustainability and environmental responsibility has driven considerable interest in utilizing renewable biomass for material development. Among such resources, cellulose—nature’s most abundant biopolymer—can be refined into various nanostructured forms, including cellulose nanocrystals, nanowhiskers, microfibrillated cellulose, fibril aggregates, and cellulose nanofibers (CNFs) [1]. Owing to their nanoscale diameter and micrometer-scale length, CNFs exhibit outstanding physicochemical properties [2]. In addition to being biodegradable and biocompatible, they offer high crystallinity, impressive mechanical strength, excellent flexibility, and notable barrier properties [3], primarily due to their large surface area and high surface reactivity [4]. These properties make CNFs attractive for diverse applications, such as reinforcement additives [5], barrier layers in biosensors [6], electromagnetic shielding [5], sustainable packaging [6,7], and usage across biomedical [8,9], cosmetic [10], pharmaceutical [11], and composite industries [5].

Conventionally, CNFs are extracted from microcrystalline cellulose [12], bleached chemical pulp [13], or cotton fibers [14] through energy-intensive mechanical processes, which hinder large-scale commercialization due to their high cost. A cost-effective alternative involves the direct transformation of lignocellulosic biomass into lignin-containing cellulose nanofibrils (LCNFs), which provides improved yield, lower production cost, and reduced environmental impact. Recent developments have highlighted deep eutectic solvents (DESs) as a green and efficient method for extracting LCNF. Compared to traditional approaches such as TEMPO-mediated oxidation or enzymatic hydrolysis, DES offers distinct advantages, including lower environmental impact, better lignin retention [15], and reduced energy consumption [16]. The residual lignin in LCNF provides enhanced thermal stability, increased hydrophobicity [17], UV resistance [18], and better compatibility with hydrophobic polymers [19]. These features render LCNF especially promising for packaging, coatings, and biocomposites [20,21].

Polyvinyl alcohol (PVA) is a synthetic, water-soluble polymer recognized for its excellent film-forming ability, biodegradability potential, and strong compatibility with natural polymers. Due to these properties, it has been widely investigated for enhancing thermal stability and mechanical performance through blending with natural materials [22,23], positioning it as a promising candidate for sustainable packaging. However, despite these advantages, PVA exhibits inherent limitations, including poor thermal resistance, high moisture absorption, and only partial biodegradability under ambient conditions [24]. To overcome these drawbacks, researchers have incorporated various bio-based nanofillers—such as nanocellulose [25], lignin [26], and chitosan [27]—into PVA matrices, resulting in significant improvements in structural integrity, thermal stability, and barrier performance.

Despite these improvements, incorporating LCNF into PVA still poses compatibility challenges. The disparity in polarity between hydrophilic PVA and relatively hydrophobic LCNF often leads to aggregation, weak interfacial adhesion, and phase separation [28], compromising the composite’s overall properties. To improve compatibility, strategies such as covalent crosslinking and metal-ion coordination have been introduced to enhance interface bonding and minimize water sensitivity [29]. Additionally, graft copolymerization has been explored to increase the hydrophilicity of lignin-rich nanofibrils, leading to better dispersion and mechanical integration within the polymer matrix [30].

More recently, innovative pretreatment methods, such as the phenolation of lignin via deep eutectic solvents, have demonstrated further improvements in LCNF–PVA film performance by strengthening interfacial cohesion and boosting composite functionality [31]. Nevertheless, a comprehensive evaluation of how various LCNF surface modification techniques affect dispersion behavior and final film properties is still lacking. Bridging this knowledge gap is essential for advancing LCNF-reinforced biodegradable materials suitable for next-generation sustainable packaging [32].

Among biomass sources, bamboo shoot shells—a major byproduct of bamboo shoot processing—represent an underutilized and abundant lignocellulosic resource [33,34,35]. In China alone, annual production of bamboo shoots reaches 3–4 million tons, generating approximately 2.4 million tons of shell residue [36]. Despite being rich in cellulose, hemicellulose, lignin, and other bioactives [37], these residues are frequently disposed of through landfilling or open burning, exacerbating environmental burdens. Their conversion into value-added materials offers a sustainable strategy for waste valorization and resource recovery.

In this study, LCNF derived from bamboo shoot shells was employed as renewable reinforcing fillers to improve the functional properties of PVA-based composite films. To address dispersion and compatibility issues within the polymer matrix, LCNF were modified via three routes: (1) hydrochloric acid (HCl) hydrolysis, which removed amorphous lignin and hemicellulose, decreased fibril diameter, and increased surface hydroxyl content for improved hydrogen bonding and dispersion [38]; (2) citric acid (CA) crosslinking, which grafted carboxylic groups onto the fibril surface and facilitated covalent ester bonding with PVA chains to strengthen interfacial adhesion [39]; and (3) a combined HCl–CA treatment, which synergistically enhanced both the physical refinement and chemical bonding, resulting in improved stress transfer and more homogeneous filler distribution. Modified LCNF were incorporated into PVA films at different concentrations, and the resulting composites were systematically characterized for thermal stability (via thermogravimetric analysis), mechanical behavior (tensile strength, elongation, and modulus), and barrier performance (oxygen and water vapor transmission rates). This investigation reveals how HCl and CA modifications synergistically enhance LCNF-PVA interactions and offers a strategy for transforming bamboo shoot shell waste into high-performance biodegradable packaging.

## 2. Materials and Methods

### 2.1. Materials

The bamboo shoot shells (dried, cellulose 81.24%, lignin 10.69%, hemicellulose 5.34%) were obtained from Jingshi Agricultural Science and Technology Co., Ltd. (Huangshan, China). Polyvinyl alcohol (PVA, MW: 88,000) was supplied by Henan Xinrui Technology Co., Ltd. (Xinxiang, China). γ-Methacryloxypropyltrimethoxysilane (KH151, powder, 97%) was purchased from Huaian Heyuan Chemical Co., Ltd. (Huaian, China). Sodium hydroxide (NaOH, ≥96%), hydrochloric acid (HCl, 37%), citric acid (99.5%), anhydrous ethanol (≥99.7%), choline chloride (≥98%), ferric chloride (FeCl_3_, ≥98%), magnesium nitrate hexahydrate (Mg(NO_3_)_2_·6H_2_O, 99.5%), and oxalic acid dihydrate (≥99%) were obtained from Sinopharm Chemical Reagent Co., Ltd. (Shanghai, China). All chemicals were of analytical grade and used without further purification.

### 2.2. Preparation and Modification of LCNF

#### 2.2.1. Preparation of LCNF

Bamboo shoot shells were first rinsed thoroughly with tap water to remove surface dirt. The cleaned shells were then air-dried at 50 °C for 24 h, chipped into small pieces (~2–3 cm), and ground into fine powder using an 80-mesh sieve. A 0.1% (*w*/*v*) sodium hydroxide solution was added to the sample at a material-to-liquid ratio of 1:5 (*w*/*w*), and the mixture was stirred at 90 °C for 2 h. After the reaction, the suspension was centrifuged at 4000 rpm for 10 min to separate the solid phase. The obtained residue was washed with deionized water multiple times until the pH became neutral and then dried at 60 °C. A deep eutectic solvent (DES) based on choline chloride, oxalic acid, and ferric chloride with a molar ratio of 1:1:0.2 was prepared by heating at 90 °C until a homogeneous solution formed. Bamboo shoot shells were mixed with DES (1:100, *w*/*v*) and heated at 90 °C for 2 h, followed by centrifugation. Residual DES was removed by dialysis in distilled water (MWCO 12–14 kDa) for 7 days until the solution became neutral. The mixture was subjected to centrifugation at 8000 rpm for 10 min, followed by drying at 50 °C The dried sample was prepared as a 0.5% (*w*/*w*) suspension and treated ultrasonically in an ice bath at 700 W for 30 min, then homogenized at 40 MPa for 5 min to produce the LCNF suspension. This method was adapted from the work of Guo et al. [40].

#### 2.2.2. Preparation and Characterization of Modified LCNF

The LCNF suspension 0.5% (*w*/*w*, 100 mL) was subjected to three distinct chemical treatments: acid hydrolysis using 0.5 mol/L HCl (100 mL), crosslinking with 0.5 mol/L citric acid (CA, 100 mL), and a combined approach involving a 1:1 (*w*/*w*) blend of 0.5 mol/L HCl (50 mL) and 0.5 mol/L CA (50 mL), resulting in final concentrations of 0.25 mol/L for both HCl and CA in the mixed solution. The suspensions were first stirred at 500 rpm for 2 h at room temperature (~25 °C) to ensure thorough mixing and reaction. Subsequently, the pH of each system was adjusted to 4 through the gradual addition of 1 mol/L NaOH solution, with continuous pH monitoring using a pH meter. The mixtures were transferred into glass beakers and dried in a vacuum oven at 60 °C for 48 h. The resulting dried samples, covered with aluminum foil, were then thermally treated in air at 130 °C for 7 h. Once cooled, the products were washed via repeated centrifugation (8000 rpm, 10 min, 3×) with deionized water until a neutral pH range (7–8) was achieved and further cleaned by sequential rinsing with acetone and methanol. Then, the samples were pre-frozen at −80 °C for 12 h before lyophilization at −50 °C under 0.1 mbar for 48 h. These processed materials were designated as LCNF-HCl, LCNF-CA, and LCNF-HCl&CA, respectively. In contrast, the untreated LCNF sample was directly lyophilized and used as the control.

The chemical structures of LCNF, LCNF-HCl, LCNF-CA, and LCNF-HCl&CA were characterized by Fourier-transform infrared (FTIR) spectroscopy. Spectral data were acquired at room temperature using a VERTEX 80v spectrometer (Bruker, Billerica, Germany) equipped with an attenuated total reflectance (ATR) module. The FTIR spectra were collected in the range of 400–4000 cm^−1^ with a spectral resolution of 4 cm^−1^ [41].

### 2.3. Preparation of the Films

LCNF, LCNF-HCl, LCNF-CA, and LCNF-HCl&CA samples were incorporated into 3 wt% PVA aqueous solutions (50 mL), which were prepared by dissolving PVA in deionized water at 95 °C under constant stirring until a clear and homogeneous solution was obtained. The fillers were added at concentrations of 5, 10, and 15 wt% relative to the dry weight of PVA, with mixing performed at 95 °C for 30 min to ensure uniform dispersion. After cooling to 50 °C, KH151 (0.5 wt% relative to PVA) was added, and the mixtures were further stirred at 50 °C for 2 h. The resulting mixtures were then cast into petri dishes (90 × 90 × 20 mm) and dried in a ventilated oven at 55 °C. Finally, the films were peeled off and stored in a desiccator containing a saturated Mg(NO_3_)_2_ solution (53% RH) at 25 °C for 72 h prior to analysis. The composite films were named PVA/LCNF, PVA/LCNF-HCl, PVA/LCNF-CA, and PVA/LCNF-HCl&CA. The neat PVA film served as the control. A schematic of the entire preparation and modification process, along with film formation, is shown in Figure 1.

### 2.4. Characterization of the Films

#### 2.4.1. Microstructure Analysis

All film samples were fractured under cryogenic conditions using liquid nitrogen to expose their cross-sections. The morphology of these cross-sections was observed by scanning electron microscopy (SEM) with a Hitachi SU8010 instrument operating at an accelerating voltage of 2.0 kV and a working distance of 15.5 mm, with images captured at a magnification of ×10,000. Prior to imaging, the samples were sputter-coated with gold (60 s) to improve surface conductivity and image quality [42].

#### 2.4.2. Light Transmittance Properties Measurement

Film transmittance was measured by a UV–Vis spectrophotometer (V-5800, Metash Instruments, Shanghai, China) at 53% RH. Samples (10 × 40 mm) were scanned from 200 to 800 nm using an empty quartz cuvette as reference [43].

#### 2.4.3. X-Ray Diffraction (XRD) Analysis

X-ray diffraction (XRD) analysis was performed on the samples using an XPert Pro MPD diffractometer (PANalytical, Almelo, The Netherlands) with Cu Kα radiation (λ = 0.154 nm) operating at 40 kV and 40 mA. Data were collected at a scanning speed of 0.1 s per step across a 2θ range from 5° to 50°. The crystallinity index (CrI) was calculated from the XRD patterns using the Segal method [44], according to Equation (1).(1)$$\mathrm{CrI}(\%)=\frac{I_{002}{-I}_{\mathrm{am}}}{I_{002}}\times 100$$
*I*_002_: (002) crystal peak intensity at 2θ ≈ 22.5°.*I_am_*: intensity of amorphous region at 2θ ≈ 18°.

#### 2.4.4. Thermogravimetric Analysis

Thermogravimetric analysis (TGA) was carried out on a NETZSCH STA 449C instrument (NETZSCH, Selb, Germany). Around 5 mg of sample was heated from 30 °C to 600 °C at a rate of 10 °C/min under a nitrogen atmosphere [45]. Thermal stability was assessed using derivative thermogravimetric (DTG) curves.

#### 2.4.5. Tensile Properties Analysis

Tensile testing of the films was conducted using a universal testing machine (MTS, E45105, China). Rectangular specimens measuring 1 × 5 cm^2^ were preconditioned at 53% relative humidity and 23 °C for 24 h. Film thickness (70–80 µm) was determined as the mean of five measurements taken at randomly selected points on each specimen with a micrometer (0.001 mm resolution; Hefei Weishi Machinery Co., Hefei, China). Samples were clamped and stretched at a crosshead speed of 5 mm/min with a 500 N load cell. Tensile strength (MPa), Young’s modulus (MPa), and elongation at break (%) were recorded at ambient temperature [46].

#### 2.4.6. Swelling and Solubility Testing

Film samples (20 mm × 20 mm) were initially weighed to determine their dry mass (M_0_). The pre-weighed films were then immersed in 30 mL of deionized water in 50 mL beakers. After 2 h of immersion at room temperature, the films were removed, and excess surface water was gently blotted using filter paper. The swelling rate was subsequently calculated using Formula (2) [47]:(2)$$\mathrm{Swelling}\;\mathrm{rate}\;(\%)=\frac{M_{t}-M_{o}}{M_{o}}\times 100$$
where *M_t_* is the mass of the swollen film, and *M*_0_ is the dry mass. The immersion time of 2 h was chosen as a fixed reference point to facilitate the comparison of swelling behavior among different film formulations. This approach was not intended for the investigation of swelling kinetics.

For the solubility test, each sample (*W*_1_) was soaked in 20 mL of distilled water at 25 °C for 24 h. The remaining insoluble material was dried at 50 °C until a constant weight (*W*_2_) was achieved [48]. The solubility rate was calculated using Formula (3):(3)$$\mathrm{Solubility}\;\mathrm{rate}\;(\%)=\frac{W_{1}-W_{2}}{W_{1}}\times 100$$

#### 2.4.7. Water Vapor and Oxygen Barrier Properties

The water vapor transmission rate (WVTR) of the films was determined using a WVTR cup (W3/060, Labthink Co., Ltd., Jinan, China). Circular specimens with an 80 mm diameter were sealed onto cup lids containing 10 mL of distilled water. Measurements were conducted via the gravimetric method at 38 °C and 50% relative humidity for 24 h [49]. WVTR was calculated using Formula (4):(4)$$\mathrm{WVTR}=\frac{\Delta \;m}{A\times t}$$
where Δ*m* is the mass difference, *A* is the test area, *t* is the time interval.

The oxygen barrier properties of the films were evaluated by measuring the oxygen transmission rate (OTR) at 23 °C and 0% relative humidity using an 8001 oxygen permeability tester (Systech Illinois, Thame, Oxfordshire, UK) [50].

## 3. Results

### 3.1. FTIR Analysis of Modified LCNF

Figure 2 presents the FTIR spectra of LCNF, LCNF-HCl, LCNF-CA, and LCNF-HCl&CA. The characteristic absorption bands of LCNF include a broad peak between 3600 and 3020 cm^−1^ attributed to O–H stretching, a band near 2900 cm^−1^ corresponding to C–H stretching, and an O–H bending vibration at 1650 cm^−1^. The peaks at 1440 cm^−1^ and 1367 cm^−1^ are assigned to –CH_2_ and C–H bending vibrations, respectively. In the 1160–1030 cm^−1^ range, C–O–C and C–O stretching vibrations appear, with a notable peak at 1060 cm^−1^ related to C–O stretching. The absorption at 895 cm^−1^ is characteristic of the β-1,4-glycosidic linkage in cellulose [51]. The FTIR results indicate that the primary functional groups of LCNF remain largely unchanged following modification. Hydrochloric acid treatment selectively hydrolyzes the amorphous regions, while citric acid esterification introduces a new absorption band at 1750 cm^−1^, confirming the formation of ester groups [52]. This demonstrates the successful incorporation of ester functionalities into the LCNF structure. The introduction of these ester groups not only modifies the surface chemistry of the nanofibers but also enhances their compatibility with hydrophobic polymer matrices by reducing interfacial tension and promoting stronger chemical interactions, thereby potentially improving interfacial adhesion in composite applications [53].

### 3.2. Microscopic Morphology

As shown in Figure 3, the cross-sectional smoothness of the modified composite films is significantly improved compared to that of the unmodified PVA/LCNF films at the same filler content. As the filler content increases from 5 to 10 wt%, both PVA/LCNF-HCl and PVA/LCNF-HCl&CA exhibit more uniform and smoother morphologies. This enhancement is mainly attributed to the selective hydrolysis of the amorphous regions of LCNF by hydrochloric acid, which not only increases the crystallinity of cellulose but also generates smaller, more uniformly distributed nanocrystals, thereby promoting better dispersion within the PVA matrix. However, all modified samples show evident particle aggregation at 15 wt%. This indicates that excessive LCNF addition disrupts uniform dispersion in the PVA matrix, ultimately compromising the overall performance of the composite films.

### 3.3. Transparency

As shown in Figure 4, all composite films maintained good light transmittance. Figure 5 illustrates that at a wavelength of 550 nm, the transmittance of all composite films was lower than that of the neat PVA film (81.22%, Figure 5a). However, the light transmittance of all composite films remained above 69%. Compared with conventional biodegradable food packaging materials, such as PLA-based or starch-based films, which typically exhibit transmittance values ranging from 60% to 75% at similar wavelengths [54,55], the developed composite films demonstrate competitive optical transparency. Notably, at different filler contents (5, 10, and 15 wt%), the modified composite films exhibited significantly higher transmittance compared to the unmodified PVA/LCNF films (Figure 5b–d). Among these, PVA/LCNF-HCl showed the highest transmittance, followed by PVA/LCNF-HCl&CA. The HCl modification effectively improved the compatibility between the filler and the matrix by enhancing the crystallinity and dispersibility of LCNF. In contrast, while CA crosslinking contributed to improved mechanical strength, the network structure it introduced generated additional light-scattering centers, leading to a relative reduction in transmittance. Furthermore, with increasing filler content, the transmittance of all samples showed a declining trend consistent with SEM observations.

### 3.4. Crystal Structure

XRD analysis (Figure 6) confirmed the semi-crystalline nature of all samples, as evidenced by the characteristic diffraction patterns of PVA. The dominant peak at 2θ = 19.5°, corresponding to the (110) crystal plane, remained consistent across all formulations, while minor peaks at 40.8° (200) and 13.1° (101) indicated preserved crystal symmetry. As summarized in Table 1, the evolution of crystallinity was influenced by both filler content and chemical modification. The neat PVA film exhibited a crystallinity of 37.50%, which increased significantly upon the incorporation of LCNF, PVA/LCNF film reaching a maximum value of 65.41% at 10 wt%. Among the modified samples, HCl modification led to the most substantial enhancement in crystallinity, achieving 79.99% at 10 wt%, representing a 23.8% improvement compared to unmodified PVA/LCNF films. This enhancement can be attributed to the hydrolysis of amorphous cellulose regions, improved interfacial adhesion between the matrix and filler, and reduced fiber diameter, which together facilitate epitaxial crystallization [56]. CA modification also improved crystallinity, though to a lesser extent (68.20 at 10 wt%), possibly due to cross-linking that restricted chain mobility. The dual modification with HCl and CA resulted in an intermediate crystallinity value of 77.27% at 10 wt%, likely due to the opposing effects of HCl-induced chain scission and CA-induced cross-linking [51]. A noticeable reduction in crystallinity was detected at a filler loading of 15 wt% in all sample groups, suggesting that excessive fiber content leads to agglomeration and limits uniform dispersion—an outcome that aligns with SEM analysis results [57].

### 3.5. Thermal Stability

Figure 7 presents the thermogravimetric analysis (TGA) profiles of the composite films, and the corresponding *T*_onset_, *T*_50_, residue at 600 °C and maximum degradation rate are summarized in Table 2. All samples exhibited an initial mass loss of approximately 3% around 150 °C, primarily due to the evaporation of residual moisture. A rapid weight loss of about 75% occurred between 200 °C and 380 °C, corresponding to the thermal degradation of PVA [58]. Beyond 380 °C, the combustion of carbonaceous residues led to a total mass loss exceeding 90% by 600 °C. The neat PVA film exhibited the lowest residue (7.91%), as shown in Figure 7a.

As shown in Figure 7b–d and Table 2, the composite films demonstrated improved thermal stability. Notably, the PVA/LCNF-HCl&CA films consistently exhibited the highest residue, T_onset_ and T_50_ values across all filler content levels (5 wt%, 10 wt%, and 15 wt%), indicating that the dual modification with HCl and CA synergistically enhanced to the improvement in thermal stability. This enhancement is likely attributed to increased crystallinity and the formation of a dense, cross-linked network structure. At 10 wt% filler content, this the PVA/LCNF-HCl&CA film achieved the highest residue (9.21%, Figure 7c), along with the highest initial decomposition temperature (T_onset_, 295.43 °C) and the temperature at 50% mass loss (T_50_, 345.13 °C). However, when the filler content increased to 15 wt% (Figure 7d), a slight decline in thermal stability was observed. The residue of the PVA/LCNF-HCl&CA composite at this level decreased to 8.82%, and both T_onset_ and T_50_ slightly dropped to 289.23 °C and 338.53 °C, respectively. This reduction may result from the agglomeration of excess nanofibers at higher loadings, which disrupts uniform dispersion and compromises the integrity of the cross-linked network, as evidenced by the increased particle clustering observed in SEM images.

The differential thermogravimetric (DTG) profiles are presented in Figure 8, with corresponding data summarized in Table 2. Notably, the degradation rate remained relatively stable across the 5–15 wt% filler content range, showing minimal variation with increasing filler loading. The neat PVA film exhibited the highest maximum degradation rate, reaching 16.17%/min (percentage mass loss per minute) in Figure 8a. All modified composite films displayed lower degradation rates compared to the unmodified PVA/LCNF film, as shown in Figure 8b–d. Among these, the dual HCl&CA treatment demonstrated the most pronounced effect, consistently yielding the lowest degradation rates across all filler contents (10.96%/min at 5 wt%, 10.81%/min at 10 wt%, 11.09%/min at 15 wt%). These results are consistent with the TGA data. Statistical analysis further confirmed that 10 wt% is the optimal filler content for achieving enhanced thermal stability in PVA-based composite films.

### 3.6. Mechanical Properties

According to Table 3, the tensile strength of the neat PVA film was measured at 27.50 MPa. Incorporating 5 wt% unmodified LCNF increased this to 30.12 MPa due to the intrinsic stiffness of nanofibers. At 10 wt%, the tensile strength further rose to 35.81 MPa, reflecting improved stress transfer through fiber networks. However, a drop to 25.86 MPa at 15 wt% suggested that fiber aggregation limited reinforcement. Surface modifications significantly enhanced strength: PVA/LCNF-HCl films improved interfacial bonding, with tensile strength values increasing from 33.69 MPa (5 wt%) to 39.68 MPa (10 wt%), before declining slightly to 35.91 MPa (15 wt%). PVA/LCNF-CA films showed moderate gains, peaking at 36.65 MPa (10 wt%). Notably, the dual-modified LCNF (HCl&CA) achieved the highest tensile strength of 49.27 MPa at 10 wt%, accompanied by a 15.1% increase in elongation at break relative to neat PVA film.

The elongation at break (%) in Table 4 followed a similar modification-dependent pattern. The neat PVA film showed an elongation of 113.92%, while unmodified LCNF reduced it across all loadings—75.58% (5 wt%), 66.71% (10 wt%), and 56.30% (15 wt%). This reduction is due to weak interfacial interactions and increased brittleness. In contrast, PVA/LCNF-HCl films reached up to 120.83% (10 wt%), though elongation declined to 84.33% at 15 wt%, likely from fiber clustering. PVA/LCNF-CA films reached a peak of 124.03% (10 wt%). The dual-modified LCNF achieved the best results, with an elongation at break of 131.14% at 10 wt%, representing a 15.1% improvement over the neat PVA film.

As shown in Table 5, Young’s modulus (MPa) of composite films varied notably with LCNF content and surface modification. The neat PVA film exhibited a modulus of 769.31 MPa, which was slightly increased by unmodified LCNF to 938.02 MPa at 10 wt% before dropping to 791.70 MPa at 15 wt% due to possible poor dispersion or fiber agglomeration. PVA/LCNF-HCl films showed improved stiffness, reaching 1002.65 MPa at 10 wt%, while PVA/LCNF-CA composites also demonstrated strong performance, peaking at 997.57 MPa. The PVA/LCNF-HCl&CA film exhibited the highest modulus of all, with 1024.10 MPa at 10 wt%, which represents a 33.1% increase compared to neat PVA, confirming its ability to enhance stiffness through synergistic interfacial bonding and optimized fiber distribution.

Among all formulations, the PVA/LCNF-HCl&CA film at 10 wt% consistently delivered the best mechanical performance, combining high tensile strength (49.27 MPa), excellent flexibility (131.14% elongation), and superior stiffness (1024.10 MPa). These values are comparable to those of conventional packaging polymers, such as polypropylene (PP, 31–43 MPa) [59], PVC (elongation 14–450%) [60], and HDPE (modulus 1100–1500 MPa) [61]. This balance of strength and flexibility underscores the composite’s potential for advanced packaging applications requiring both durability and formability. In particular, the high elongation at break enhances its suitability for flexible packaging applications that require considerable deformability, such as tray lamination for irregularly shaped fruits and vegetables.

### 3.7. Swelling and Solubility

As shown in Figure 9a, all composite films exhibited significantly lower swelling ratios compared to neat PVA, suggesting that the incorporation of LCNF enhanced the material’s resistance to swelling. The swelling behavior showed a distinct dependence on LCNF content. At 5 wt%, the PVA/LCNF film displayed the highest swelling ratio of 20.492%, likely due to hydrogen bonding between the hydroxyl groups of LCNF and water molecules. However, increasing the LCNF content to 10 wt% led to a notable reduction in swelling ratio to 17.53%, indicating that higher filler loading promoted greater crosslinking density within the polymer network. The dual-modified PVA/LCNF-HCl&CA films demonstrated superior anti-swelling performance, exhibiting the lowest swelling ratio of 17.21% at 5 wt% and further reduced to 14.26% at 10 wt%. However, when the filler content reached 15 wt%, excessive fiber content led to aggregation and disrupted the polymer network. As a result, all samples exhibited a notable increase in swelling ratio. SEM analysis confirmed that fiber aggregation at this concentration induced microstructural defects, reducing the integrity of the network and thus increasing water absorption.

The dissolution data (Figure 9b) indicated that all composite films exhibited significantly lower solubility compared to the neat PVA film. The dissolution behavior of the PVA/LCNF films followed a concentration-dependent trend, showing an initial decrease followed by an increase: from 6.36% at 5 wt% to a minimum of 5.15 at 10 wt%, then rising to 5.70% at 15 wt%. In contrast, the PVA/LCNF-HCl&CA films exhibited the lowest solubility across all concentrations, reaching a minimum of 3.21% at 10 wt%. The enhanced anti-dissolution performance can be attributed to two synergistic mechanisms. Firstly, HCl treatment facilitated the selective dissociation of hydroxyl groups on the LCNF surface, resulting in the formation of a compact nanofiber network, as confirmed by SEM images revealing a smoother and more compact cross-sectional morphology of the modified composite films. Secondly, ester bond formation through citric acid crosslinking significantly restricted polymer chain mobility. This superior structural stability was further supported by thermogravimetric analysis (TGA), with the modified composite films showing an increased residue of 9.21% at 600 °C. The dissolution rates of all film types increased again at 15 wt%, which is attributed to network loosening caused by excessive fiber entanglement at high concentrations.

### 3.8. Barrier Performance Characterization

Figure 10a reveals that unmodified PVA films demonstrated maximal water vapor permeability among all tested samples, with a measured WVTR of 0.85 g/(m^2^·24 h). With the incorporation of LCNF, the WVTR showed a decreasing trend. At 5 wt%, the WVTR of the PVA/LCNF film decreased to 0.80 g/(m^2^·24 h), while PVA/LCNF-HCl, PVA/LCNF-CA, and PVA/LCNF-HCl&CA exhibited further reductions to 0.77, 0.78, and 0.76 g/(m^2^·24 h), respectively, indicating that modified LCNF more effectively enhanced the water vapor barrier properties. At 10 wt%, these values continued to decline, reaching 0.76, 0.72, 0.73, and 0.69 g/(m^2^·24 h), respectively. Remarkably, the PVA/LCNF-HCl&CA film demonstrated the lowest WVTR, showing an 18.8% decrease relative to the neat PVA film. This indicates the synergistic enhancement of water vapor barrier properties achieved through combined HCl and citric acid modification. However, at 15 wt%, a slight increase was observed across all composite films, likely due to LCNF aggregation and the resulting structural defects compromising the barrier function.

As shown in Figure 10b, the OTR of the neat PVA film was 10.22 cc/(m^2^·day). Upon incorporation of LCNF, the OTR decreased to 9.70 cc/(m^2^·day) at 5 wt%, further dropped to 9.28 cc/(m^2^·day) at 10 wt%, and slightly increased to 9.84 cc/(m^2^·day) at 15 wt%. This increase is attributed to fiber agglomeration or microstructural defects. In addition, surface modification of LCNF further improved the oxygen barrier performance. At 10 wt%, the PVA/LCNF-HCl&CA film achieved the lowest OTR of 8.32 cc/(m^2^·day), representing an 18.6% reduction compared to the neat PVA film, followed by PVA/LCNF-HCl (8.76) and PVA/LCNF-CA (8.78 cc/(m^2^·day)). These improvements are attributed to enhanced interfacial compatibility and more uniform filler dispersion resulting from chemical modifications. At 15 wt%, OTR values increased in all modified films, with PVA/LCNF-HCl&CA at 9.24 cc/(m^2^·day), suggesting that excessive filler content may disrupt the dense network structure. Overall, the results indicate that the PVA/LCNF-HCl&CA composite film at a 10 wt% loading achieves the most effective enhancement in both WVTR and OTR. These trends align with SEM observations and confirm that 10 wt% is the optimal loading.

From a practical standpoint, although the oxygen transmission rate (OTR, 8.32 cc/(m^2^·day)) of the PVA/LCNF-HCl&CA composite film does not match the ultra-high barrier performance of EVOH (0.325 cc/(m^2^·day)) [62], it is sufficient for preserving low- to medium-respiration fruits and vegetables such as apples and carrots. The film also exhibits an excellent water vapor barrier (WVTR, 0.69 g/(m^2^·24 h)); however, its swelling rate remains relatively high (14.26%). Therefore, additional treatment may be necessary to ensure long-term stability in high-humidity environments, or its use may be better suited for short-term packaging applications.

## 4. Conclusions

This work successfully demonstrates the enhancement of PVA-based biodegradable films through the incorporation of chemically modified LCNF derived from bamboo shoot shells. Dual modification with HCl and CA proved effective in improving mechanical properties, thermal stability, and barrier performance while simultaneously reducing water sensitivity. These improvements are attributed to the synergistic effects of enhanced interfacial adhesion and uniform dispersion enabled by dual chemical modifications. Importantly, this approach not only contributes to the valorization of agricultural waste but also aligns with the growing demand for sustainable and eco-friendly packaging materials. Future studies may focus on optimizing large-scale production processes and investigating the biodegradation behavior of these films under real-world environmental conditions.

## Figures and Tables

**Figure 1 polymers-17-01571-f001:**
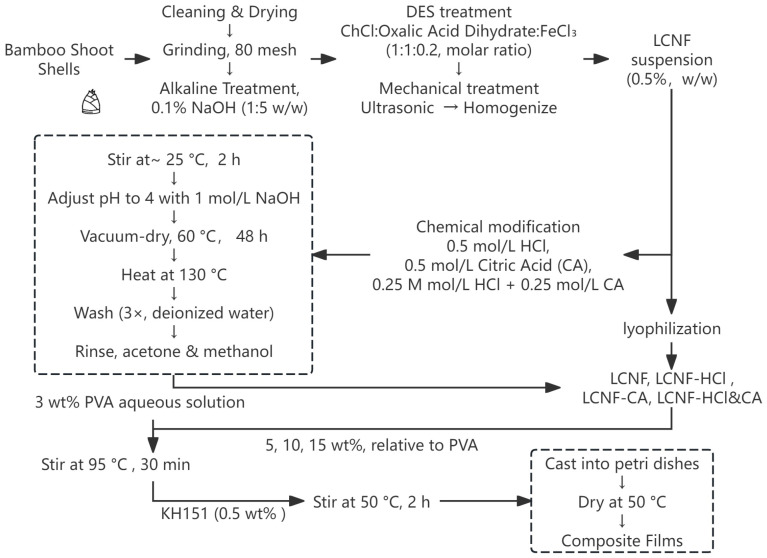
Process flow of LCNF preparation, chemical modification, and composite film formation.

**Figure 2 polymers-17-01571-f002:**
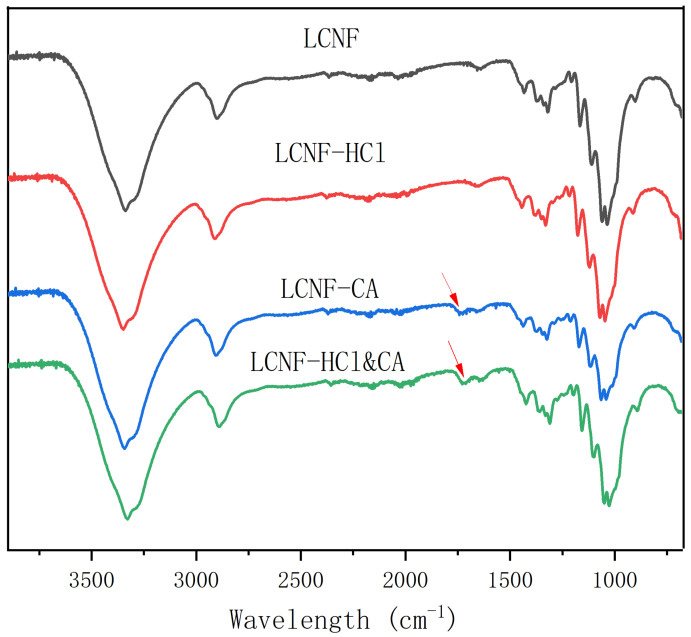
FTIR spectra of LCNF, LCNF-HCl, LCNF-CA, and LCNF-HCl&CA.

**Figure 3 polymers-17-01571-f003:**
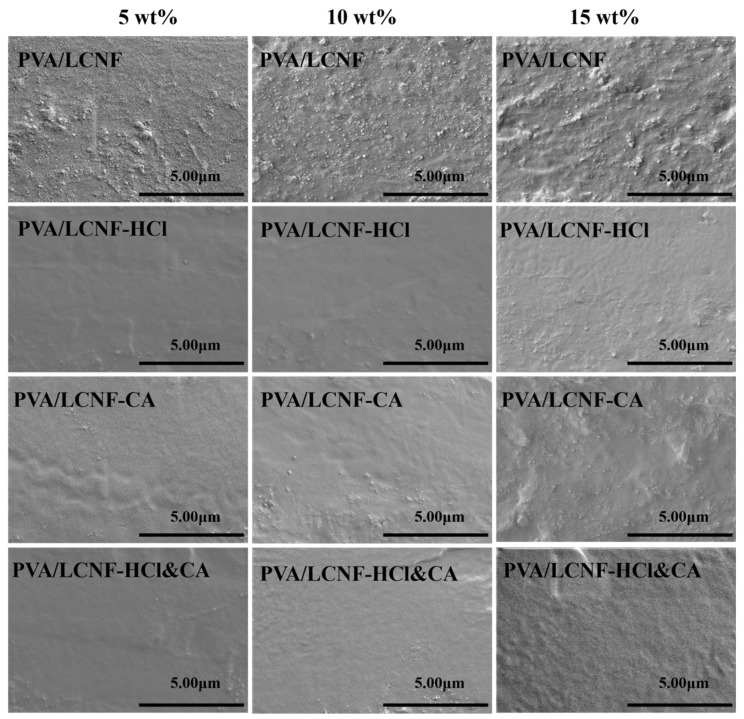
SEM images of the cross-sectional morphologies of the composite films.

**Figure 4 polymers-17-01571-f004:**
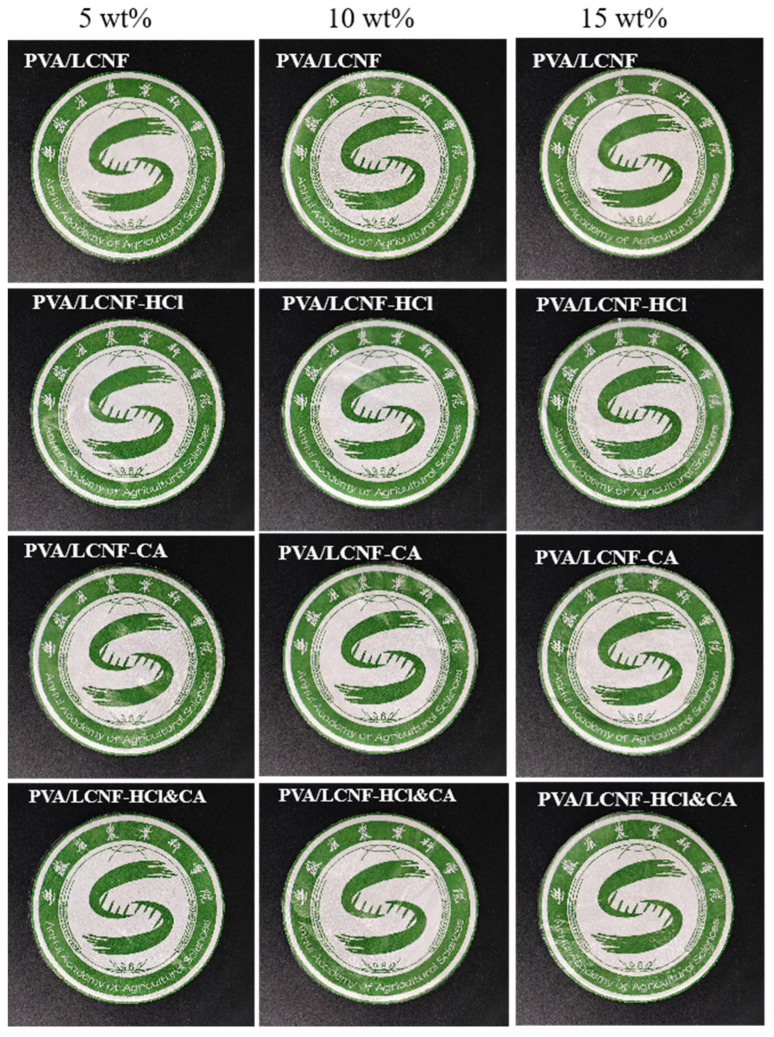
Optical transmittance images of the composite films.

**Figure 5 polymers-17-01571-f005:**
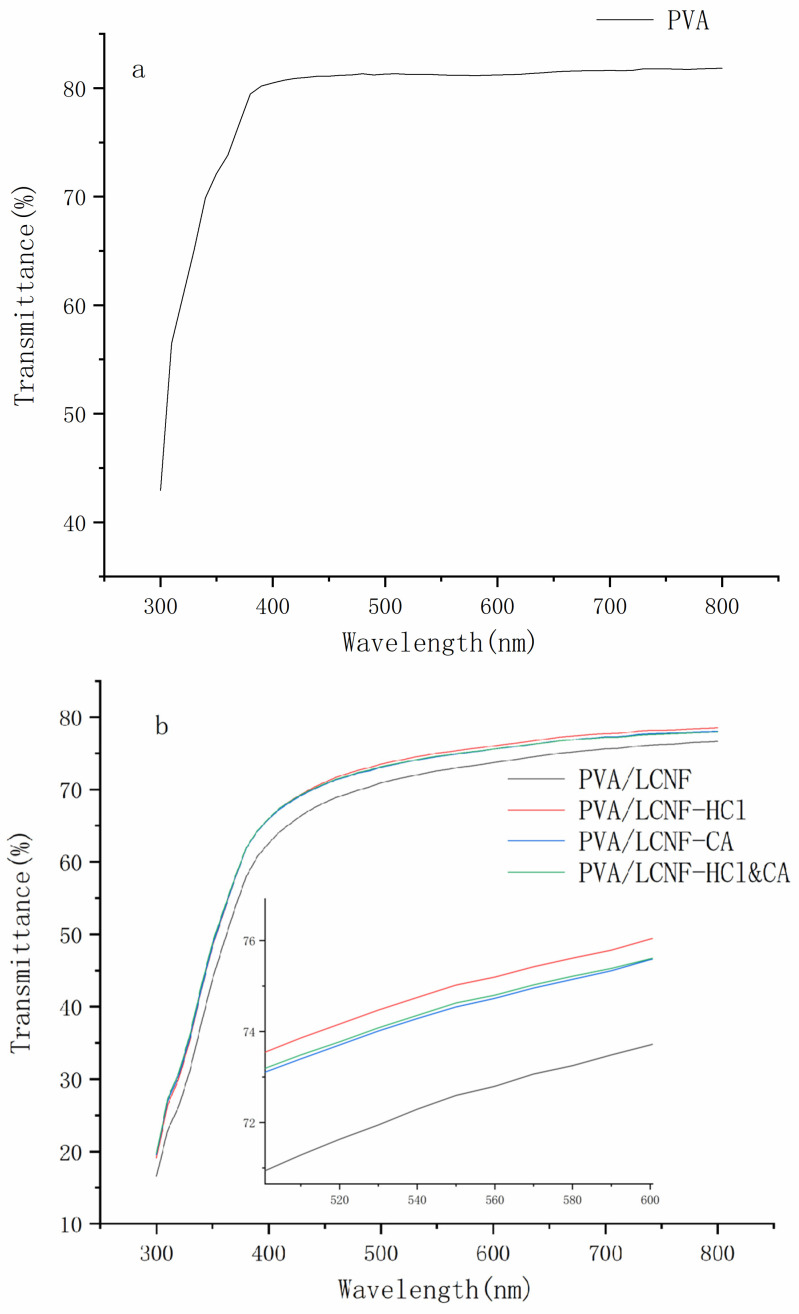
Light transmittance of the composite films ((**a**): neat PVA; (**b**): PVA/LCNF-5 wt%; (**c**): PVA/LCNF-10 wt%; (**d**): PVA/LCNF-15 wt%).

**Figure 6 polymers-17-01571-f006:**
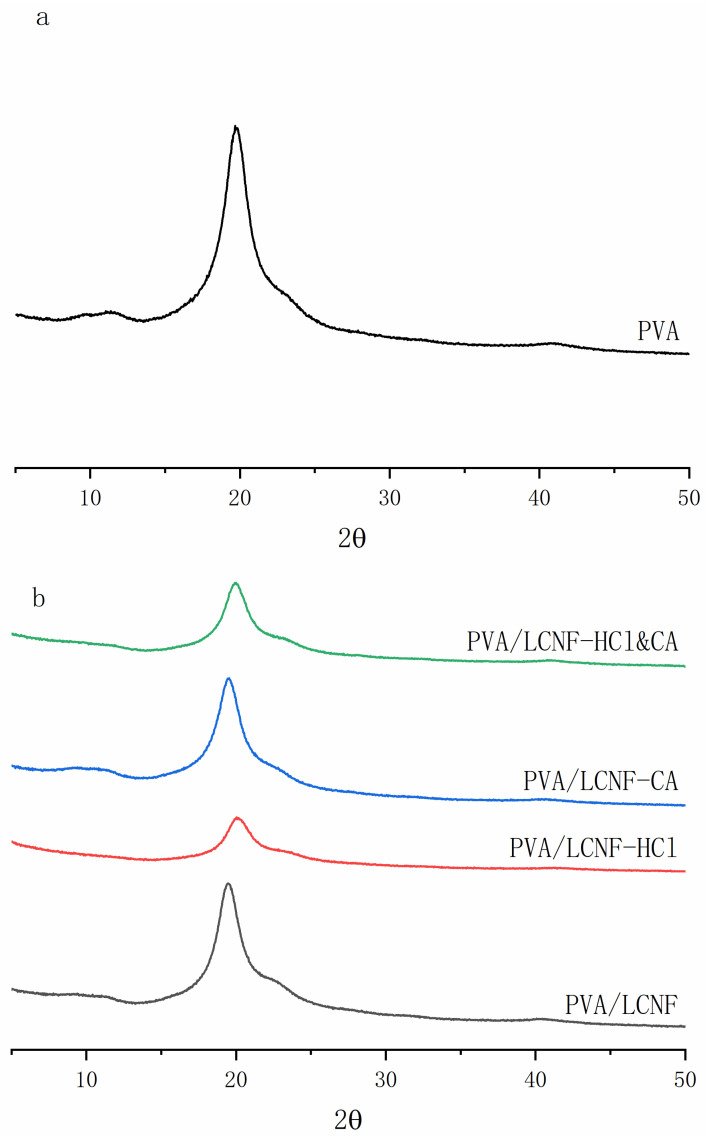
XRD patterns of the composite films ((**a**): neat PVA; (**b**): PVA/LCNF-5 wt%; (**c**): PVA/LCNF-10 wt%; (**d**): PVA/LCNF-15 wt%).

**Figure 7 polymers-17-01571-f007:**
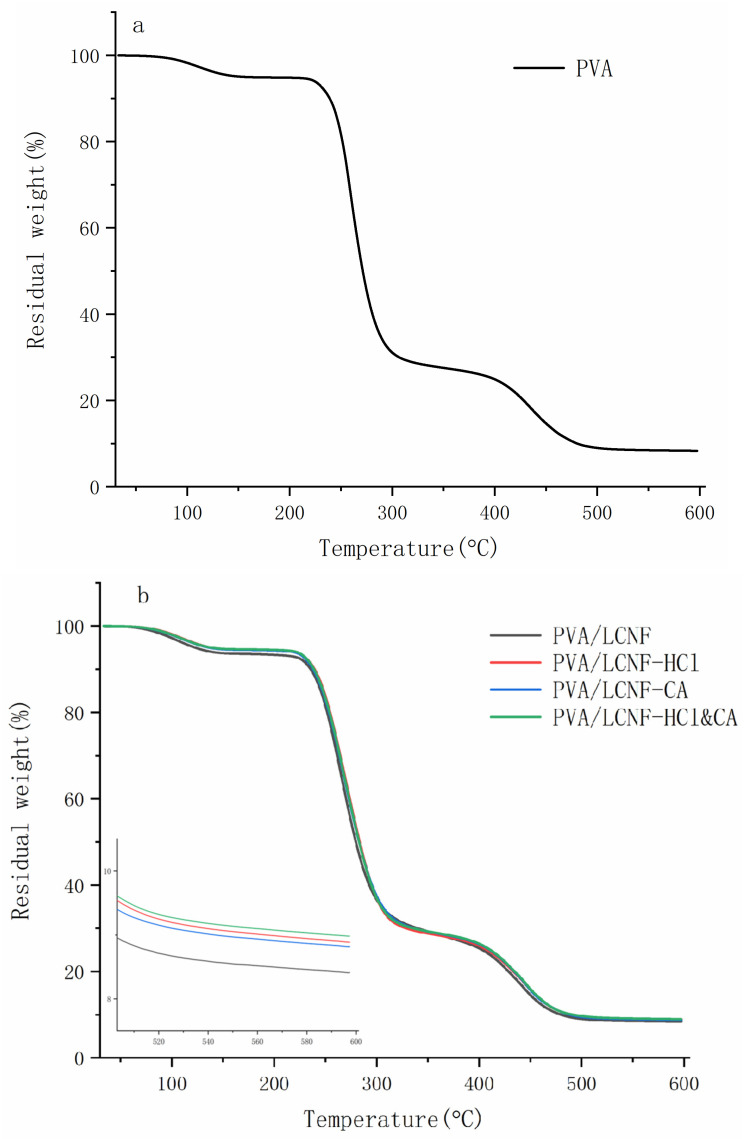
TGA profiles of the composite films ((**a**): neat PVA; (**b**): PVA/LCNF-5 wt%; (**c**): PVA/LCNF-10 wt%; (**d**): PVA/LCNF-15 wt%).

**Figure 8 polymers-17-01571-f008:**
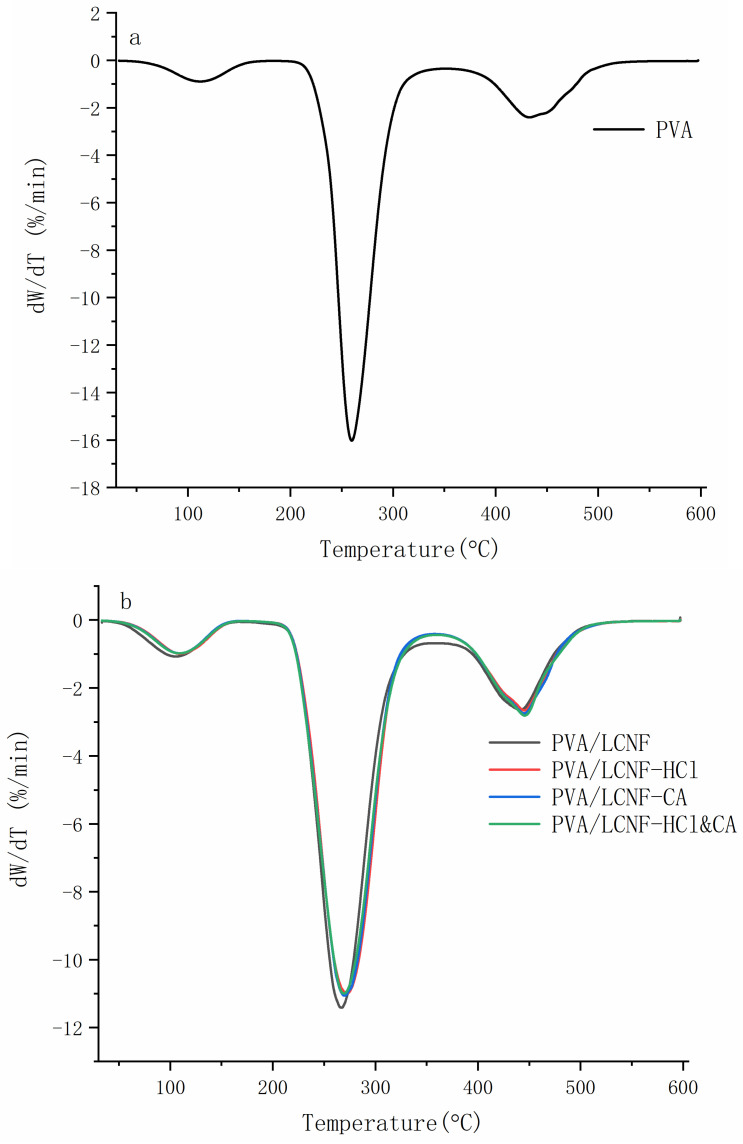
DTG profiles of the composite films ((**a**): neat PVA; (**b**): PVA/LCNF-5 wt%; (**c**): PVA/LCNF-10 wt%; (**d**): PVA/LCNF-15 wt%).

**Figure 9 polymers-17-01571-f009:**
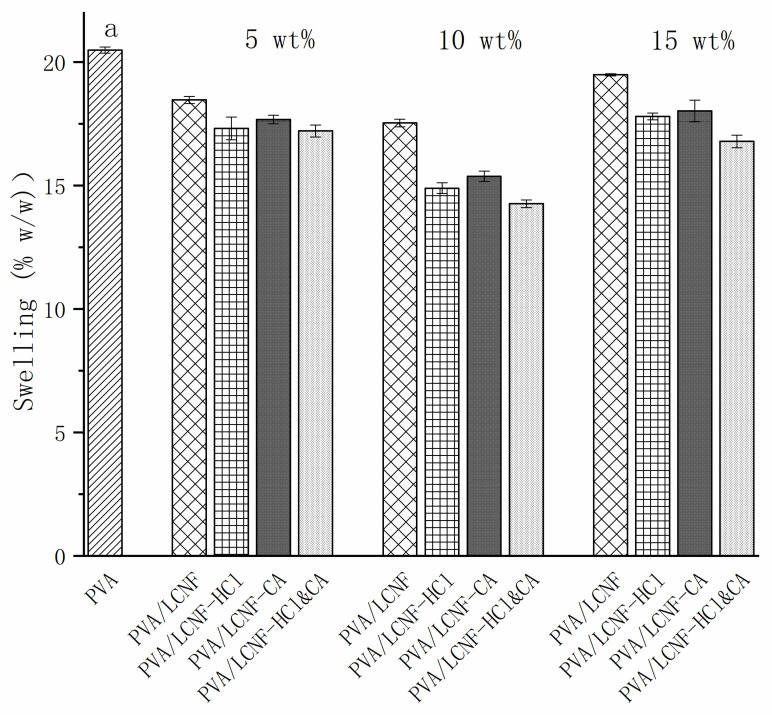
Swelling rate (**a**) and solubility rate (**b**) of the composite films.

**Figure 10 polymers-17-01571-f010:**
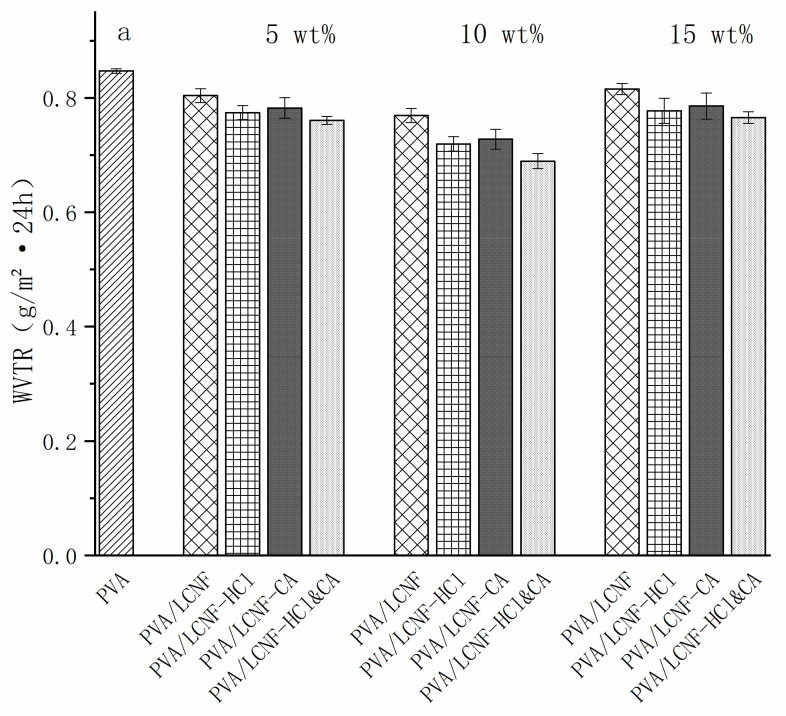
WVTR (**a**) and OTR (**b**) of the composite films.

**Table 1 polymers-17-01571-t001:** Crystallinity (%) of composite films with varying filler contents.

Sample	0%	5 wt%	10 wt%	15 wt%
PVA	37.50 ± 0.31 ^a^	-	-	-
PVA/LCNF	-	64.42 ± 0.51 ^b^	65.41 ± 0.62 ^c^	62.45 ± 0.48 ^d^
PVA/LCNF-HCl	-	69.23 ± 0.35 ^e^	79.99 ± 0.81 ^f^	75.61 ± 0.29 ^g^
PVA/LCNF-CA	-	65.02 ± 0.42 ^b^	68.20 ± 0.56 ^h^	64.24 ± 0.57 ^d^
PVA/LCNF-HCl&CA	-	68.50 ± 0.60 ^e^	77.27 ± 0.72 ^i^	72.50 ± 0.64 ^j^

Data are presented as mean ± standard deviation (n = 3). Statistically significant differences (*p* < 0.05) within each column are indicated by different superscript letters (a–j), based on two-way ANOVA followed by Tukey’s HSD post hoc analysis. The 0 wt% PVA control was not included in the statistical comparison among the composite samples.

**Table 2 polymers-17-01571-t002:** Thermal characteristics from TGA and DTG of composites films.

Filler Content (wt%)	Sample	*T*_onset_ (°C)	*T*_50_ (°C)	Residue (%)	Max Degradation Rate (%/min)
0	PVA	265.4 ± 1.23 ^a^	315.22 ± 1.81 ^a^	7.91 ± 0.17 ^a^	16.17 ± 0.51 ^d^
5	PVA/LCNF	275.12 ± 1.51 ^b^	325.23 ± 2.45 ^b^	8.41 ± 0.28 ^ab^	11.52 ± 0.45 ^c^
	PVA/LCNF-HCl	281.31 ± 1.24 ^c^	331.63 ± 1.67 ^c^	8.88 ± 0.23 ^bc^	10.97 ± 0.56 ^b^
	PVA/LCNF-CA	290.53 ± 1.62 ^d^	340.83 ± 2.75 ^d^	8.84 ± 0.31 ^bc^	10.99 ± 0.54 ^b^
	PVA/LCNF-HCl&CA	292.73 ± 2.03 ^d^	342.93 ± 1.73 ^d^	8.98 ± 0.22 ^c^	10.96 ± 0.61 ^b^
10	PVA/LCNF	280.83 ± 1.89 ^c^	330.43 ± 1.95 ^c^	8.45 ± 0.21 ^ab^	11.41 ± 0.35 ^c^
	PVA/LCNF-HCl	285.23 ± 2.06 ^d^	335.03 ± 2.04 ^d^	9.05 ± 0.27 ^cd^	10.93 ± 0.39 ^b^
	PVA/LCNF-CA	286.73 ± 1.73 ^d^	336.53 ± 1.74 ^d^	8.63 ± 0.20 ^bc^	11.38 ± 0.56 ^c^
	PVA/LCNF-HCl&CA	295.43 ± 1.81 ^e^	345.13 ± 1.88 ^e^	9.21 ± 0.17 ^d^	10.81 ± 0.58 ^a^
15	PVA/LCNF	272.53 ± 1.90 ^b^	320.83 ± 1.65 ^b^	8.1 ± 0.24 ^a^	11.10 ± 0.71 ^c^
	PVA/LCNF-HCl	278.63 ± 1.42 ^c^	327.33 ± 1.83 ^c^	8.65 ± 0.26 ^bc^	11.39 ± 0.41 ^c^
	PVA/LCNF-CA	283.93 ± 1.23 ^d^	332.73 ± 1.76 ^d^	8.29 ± 0.32 ^ab^	11.46 ± 0.26 ^c^
	PVA/LCNF-HCl&CA	289.23 ± 1.58 ^e^	338.53 ± 1.47 ^e^	8.82 ± 0.14 ^c^	11.09 ± 0.31 ^c^

*T*_onset_: Initial decomposition temperature, determined at 5% mass loss. *T*_50_: Temperature at 50% mass loss. Data expressed as mean ± standard deviation (n ≥ 3). Different superscript letters (a–e) within each column indicate statistically significant differences (*p* < 0.05) determined by two-way ANOVA with Tukey’s post hoc test.

**Table 3 polymers-17-01571-t003:** Tensile strength (MPa) of the composite films.

Sample	0%	5 wt%	10 wt%	15 wt%
PVA	27.50 ± 0.58 ^d^	-	-	-
PVA/LCNF	-	30.12 ± 2.61 ^c^	35.81 ± 2.47 ^b^	25.86 ± 1.27 ^d^
PVA/LCNF-HCl	-	33.69 ± 0.47 ^b^	39.68 ± 1.36 ^a^	35.91 ± 1.59 ^b^
PVA/LCNF-CA	-	32.35 ± 1.21 ^b^	36.65 ± 1.24 ^b^	34.32 ± 1.27 ^c^
PVA/LCNF-HCl&CA	-	37.52 ± 1.45 ^a^	49.27 ± 1.54 ^a^*	35.46 ± 1.28 ^b^

Data presented as mean ± standard deviation (n ≥ 3). Different lowercase superscript letters (a, b, c, d) indicate statistically significant differences between groups (*p* < 0.05) by two-way ANOVA with Tukey’s post hoc test. The a* symbol denotes the group with significantly highest value (*p* < 0.001 vs. all other groups).

**Table 4 polymers-17-01571-t004:** Elongation at break (%) of the composite films.

Sample	0%	5 wt%	10 wt%	15 wt%
PVA	113.92 ± 1.10 ^a^	-	-	-
PVA/LCNF	-	75.58 ± 2.96 ^b^	66.71 ± 1.69 ^c^	56.30 ± 2.41 ^d^
PVA/LCNF-HCl	-	118.32 ± 0.89 ^a^	120.83 ± 2.58 ^a^	84.33 ± 3.02 ^e^
PVA/LCNF-CA	-	117.81 ± 2.30 ^a^	124.03 ± 2.36 ^f^	82.76 ± 2.41 ^e^
PVA/LCNF-HCl&CA	-	124.18 ± 2.76 ^f^	131.14 ± 2.93 ^g^	96.88 ± 2.43 ^h^

Data are presented as mean ± standard deviation (n ≥ 3). Different lowercase superscript letters (a–h) indicate statistically significant differences (*p* < 0.05) as determined by two-way ANOVA with Tukey’s post hoc test.

**Table 5 polymers-17-01571-t005:** Young’s modulus (MPa) of the composite films.

Sample	0%	5 wt%	10 wt%	15 wt%
PVA	769.31 ± 5.81 ^a^	-	-	-
PVA/LCNF	-	789.30 ± 26.12 ^b^	938.02 ± 24.70 ^c^	791.70 ± 12.75 ^b^
PVA/LCNF-HCl	-	864.30 ± 4.74 ^d^	1002.65 ± 13.66 ^e^	971.18 ± 15.91 ^ef^
PVA/LCNF-CA	-	833.33 ± 12.15 ^d^	997.57 ± 12.49 ^e^	879.17 ± 13.73 ^f^
PVA/LCNF-HCl&CA	-	944.10 ± 14.56 ^e^	1024.10 ± 15.43 ^e^	992.89 ± 12.89 ^e^

Data are presented as mean ± standard deviation (n ≥ 3). Different lowercase superscript letters (a–f) indicate statistically significant differences (*p* < 0.05) as determined by two-way ANOVA with Tukey’s post hoc test. Shared letters denote no significant difference between groups.

## Data Availability

All data generated or analyzed during this study are included in this published article.

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
