# Peer review of "Enhancement of Polyvinyl Alcohol-Based Films by Chemically Modified Lignocellulosic Nanofibers Derived from Bamboo Shoot Shells"

_polymers, 2025, doi:10.3390/polym17111571_

Round 1
Reviewer 1 Report
Comments and Suggestions for Authors
The manuscript presents a well-executed study on the development of biodegradable composite films by reinforcing polyvinyl alcohol (PVA) with lignocellulosic nanofibers (LCNF) derived from bamboo shoot shells. The approach of dual modification via HCl hydrolysis and citric acid crosslinking is innovative and demonstrates significant improvements in mechanical, thermal, and barrier properties of the PVA-based films. The work is timely, addressing the urgent need for sustainable packaging alternatives and effective utilization of agricultural waste.
Suggestions for Improvement:
-
Comparison with Commercial Materials:
While the study demonstrates performance enhancement over neat PVA, it would benefit from a more detailed comparison with conventional plastic packaging materials (e.g., LDPE, PET) to highlight the real-world competitiveness of the composite film in terms of mechanical strength, barrier properties. -
Biodegradability Testing:
Although the film is presented as biodegradable, no experimental biodegradation or compostability data is provided. Including such tests (e.g., soil burial, enzymatic degradation, or industrial composting) would strengthen the environmental claims.

Author Response
Response to Reviewers
We sincerely thank the four reviewers for their thorough and constructive comments on our manuscript. We have carefully addressed each comment point by point and made the corresponding revisions, which are marked in red in the revised manuscript. If any modifications are still inadequate or unclear, we would greatly appreciate further guidance.
Comments1: 1. Lines 42–46:
The sentence listing numerous applications of cellulose nanofibers is broad and ambitious in
scope. However, only a single reference is provided ([5]) to support this wide array of
applications. This is insufficient and does not adequately substantiate the claims made.
Multiple citations covering specific applications (e.g., biomedical, pharmaceutical, and
packaging sectors) should be added to lend credibility and show comprehensive engagement
with the literature.
Lines 52–54:
The statement about the properties imparted by residual lignin in LCNF—such as reduced
hydrophilicity, increased thermal stability, hydrophobicity, and UV resistance—is entirely
uncited. This paragraph discusses key functional attributes of LCNF but lacks any supporting
literature. At least two or more references are necessary here to validate these claims.
Response1:We appreciate your valuable feedback and have carefully revised the manuscript to ensure proper citation and clarity. The relevant sections have been rewritten as follows:
“In addition to being biodegradable and biocompatible, they offer high crystallinity, impressive mechanical strength, excellent flexibility, and notable barrier properties [3], primarily due to their large surface area and high surface reactivity [4]. These properties make CNFs attractive for diverse applications, such as reinforcement additives [5], barrier layers in biosensors [6], electromagnetic shielding [5], sustainable packaging [6,7], and usage across biomedical [8,9], cosmetic [10], pharmaceutical [11], and composite industries [5].”
“Conventionally, CNFs are extracted from microcrystalline cellulose [12], bleached chemical pulp [13], or cotton fibers [14] through energy-intensive mechanical processes, which hinder large-scale commercialization due to their high cost. A cost-effective alternative involves the direct transformation of lignocellulosic biomass into lignin-containing cellulose nanofibrils (LCNF), which provides improved yield, lower production cost, and reduced environmental impact. Recent developments have highlighted deep eutectic solvents (DES) as a green and efficient method for extracting LCNF. Compared to traditional approaches such as TEMPO-mediated oxidation or enzymatic hydrolysis, DES offers distinct advantages, including lower environmental impact, better lignin retention [15], and reduced energy consumption [16]. The residual lignin in LCNF provides enhanced thermal stability [15], increased hydrophobicity [17], UV resistance [18], and better compatibility with hydrophobic polymers [19]. These features render LCNF especially promising for packaging, coatings, and biocomposites [20,21].
Comments1:2. Structure Issues:
The paragraph covering general information about LCNF (lines 47–66) is quite lengthy and
contains many general, well-established facts. In contrast, the section addressing the actual
challenges of integrating LCNF with PVA (lines 74–79)—arguably the core motivation for
the study—is underdeveloped and lacking in detail. This creates an imbalance in the narrative
and fails to highlight the novelty of the research effectively. The introduction of this part
should be restructured for clarity and emphasis.
Response1:We sincerely appreciate your valuable feedback. In response to your comments, we have carefully revised the manuscript by streamlining certain sections while incorporating the suggested additions. The updated text now reads:
“Conventionally, CNFs are extracted from microcrystalline cellulose [12], bleached chemical pulp [13], or cotton fibers [14] through energy-intensive mechanical processes, which hinder large-scale commercialization due to their high cost. A cost-effective alternative involves the direct transformation of lignocellulosic biomass into lignin-containing cellulose nanofibrils (LCNF), which provides improved yield, lower production cost, and reduced environmental impact. Recent developments have highlighted deep eutectic solvents (DES) as a green and efficient method for extracting LCNF. Compared to traditional approaches such as TEMPO-mediated oxidation or enzymatic hydrolysis, DES offers distinct advantages, including lower environmental impact, better lignin retention [15], and reduced energy consumption [16]. The residual lignin in LCNF provides enhanced thermal stability [15], increased hydrophobicity [17], UV resistance [18], and better compatibility with hydrophobic polymers [19]. These features render LCNF especially promising for packaging, coatings, and biocomposites [20,21].
Polyvinyl alcohol (PVA), a synthetic polymer known for its film-forming capacity, biocompatibility, and resistance to organic solvents [23], is widely considered a candidate for sustainable packaging. Nevertheless, it suffers from inherent limitations, such as poor thermal resistance, high moisture uptake, and partial biodegradability under ambient conditions [24]. To address these drawbacks, various bio-based nanofillers—such as nanocellulose [25], lignin [26], and chitosan [27]—have been incorporated into PVA matrices, resulting in enhanced structural, thermal, and barrier performance.
Despite these improvements, incorporating LCNF into PVA still poses compatibility challenges. The disparity in polarity between hydrophilic PVA and relatively hydrophobic LCNF often leads to aggregation, weak interface adhesion, and phase separation [28], compromising the composite’s overall properties. To improve compatibility, strategies such as covalent crosslinking and metal-ion coordination have been introduced to enhance interface bonding and minimize water sensitivity [29]. Additionally, graft copolymerization has been explored to increase the hydrophilicity of lignin-rich nanofibrils, leading to better dispersion and mechanical integration within the polymer matrix [30].
More recently, innovative pretreatment methods, such as phenolation of lignin via deep eutectic solvents, have demonstrated further improvements in LCNF–PVA film performance by strengthening interfacial cohesion and boosting composite functionality [31]. Nevertheless, a comprehensive evaluation of how various LCNF surface modification techniques affect dispersion behavior and final film properties is still lacking. Bridging this knowledge gap is essential for advancing LCNF-reinforced biodegradable materials suitable for next-generation sustainable packaging.
Among biomass sources, bamboo shoot shells—a major byproduct of bamboo shoot processing—represent an underutilized and abundant lignocellulosic resource. In China alone, annual production of bamboo shoots reaches 3–4 million tons, generating approximately 2.4 million tons of shell residue [33–35]. Despite being rich in cellulose, hemicellulose, lignin, and other bioactives [37], these residues are frequently disposed of through landfilling or open burning, exacerbating environmental burdens. Their conversion into value-added materials offers a sustainable strategy for waste valorization and resource recovery. ”
Comments1:3. In the paragraph discussing bamboo shoot shells (BSS), the authors fail to provide
quantitative data to support their use as a biomass source. While bamboo is known for its
rapid growth and high annual yield, the specific quantity of shell waste produced
annually—and what proportion of the plant this constitutes—is not addressed. Without this
information, the sustainability and practicality of using BSS as a lignocellulosic feedstock
remain speculative.
Response1:We sincerely appreciate your constructive comments. In response, we have incorporated the following enhanced discussion regarding bamboo shoot shell valorization:
“Bamboo shoot shells, a byproduct of bamboo shoot processing, represent a significant yet underutilized lignocellulosic resource with strong potential for value-added applications[32]. China, as the world’s largest producer of bamboo shoots, generates 3–4 million tons of fresh shoots annually, resulting in approximately 1.2–2.4 million tons of shell waste—accounting for 40–60% of the total biomass [33,34]. In Southeast Asia, 0.5–0.8 million tons of these shells are produced each year[35]. At the industrial scale, processing 10,000 tons of fresh bamboo shoots typically yields 4,000–6,000 tons of residual shells, and due to higher processing efficiencies, this ratio may reach up to 60% of the raw material mass[36]. Despite their abundance, bamboo shoot shells are frequently discarded through landfilling, open burning, or natural decomposition, practices that contribute to environmental pollution and resource waste. Recent studies have shown that these shells are rich in cellulose, hemicellulose, lignin, and bioactive compounds such as phenolics and flavonoids [37], highlighting their potential for biorefinery and material applications. However, the lack of efficient processing technologies and limited awareness of their value have hindered their utilization. Addressing this issue calls for innovative strategies to valorize bamboo shoot shells and transform them from agricultural waste into sustainable bioresources.”
Comments1:4. In the last paragraph of the introduction (around line 93), the authors outline three
strategies to address interfacial compatibility and dispersion challenges in LCNF–PVA
composites. However, they do not justify why these particular strategies were chosen or
how they are theoretically or empirically expected to resolve the known issues. Could the
author provide a brief explanation or cite prior studies showing how each strategy
contributes to improving interfacial bonding or dispersion?
Response1:We sincerely appreciate your valuable feedback. In response to your suggestions, we have carefully revised this section and incorporated the recommended references. The updated text now reads:
“In this study, LCNF derived from bamboo shoot shells were utilized as sustainable reinforcing agents to enhance the performance of PVA-based composite films. To address interfacial compatibility and dispersion challenges within the PVA matrix, LCNF were chemically modified through three strategies: hydrochloric acid (HCl) hydrolysis to remove amorphous lignin and hemicellulose, reduce fibril diameter, and expose abundant surface –OH groups for improved hydrogen bonding and dispersion [37]; citric acid (CA) crosslinking to graft carboxyl moieties onto the fibril surface and form covalent ester “bridges” with PVA chains, thus strengthening interfacial adhesion [38]; and a dual HCl&CA treatment combining the physical benefits of acid‐induced size reduction with the chemical reinforcement of ester crosslinks, yielding synergistically enhanced stress transfer and uniform filler distribution [49]. The modified LCNF were incorporated into PVA films at different loadings, and the resulting composites were evaluated for their thermal stability (via thermogravimetric analysis), mechanical properties (tensile strength, elongation at break, Young’s modulus), and barrier performance (water vapor and oxygen transmission rates). This investigation reveals how HCl and CA modifications synergistically enhance LCNF-PVA interactions and offers a strategy to transforming bamboo shoot shell waste into high-performance biodegradable packaging.”
Comments1:5. In the materials and methods part, the purity of sodium hydroxide, choline chloride are
missing. The symbol “Mt” and “Mo” is not similar to the one in line 189.
Response1:We sincerely appreciate the reviewer's careful reading and valuable suggestions. In response to the comments regarding the Materials and Methods section, we have made the following revisions:
Added the purity specifications for all chemicals:
Sodium hydroxide (NaOH, ≥96%)
Choline chloride (≥98%)
Corrected the symbols "Mt" and "Mo" to ensure consistency with Equation (2) in line 189, changing them to "Mₜ" and "M₀" respectively to maintain uniform notation throughout the manuscript.
Comments1:6. Figure 1 currently focuses only on the synthesis of LCNF, which is not a novel
contribution on its own. Since the paper emphasizes the entire process — from raw material
to film formation — the figure should reflect that full workflow. I recommend updating
Figure 1 to show the complete methodology.
Response1:
We sincerely appreciate the reviewer's insightful suggestion regarding Figure 1. In response to your valuable feedback, we have comprehensively revised the figure to better reflect the complete workflow of our study. The updated Figure 1 now includes:
The full process from bamboo shoot shell preparation to final film formation
All key steps: raw material processing, LCNF extraction, chemical modification, and composite film fabrication
Clear visual representation of the integrated methodology
This revision better aligns with the paper's emphasis on the complete value chain from agricultural waste to functional packaging material. We believe the enhanced figure now more effectively communicates our holistic approach and represents a significant improvement to the manuscript.
Comments1:7. Mising Y axis name on the Figure 2. Figure 4 is not clear, all the image looks the same.
Response1:We sincerely appreciate the reviewer's careful examination of our figures and their valuable suggestions for improvement. We have addressed both concerns as follows:
For Figure 2 (FTIR spectra):
We have maintained the current presentation format without Y-axis labels, as this is the conventional approach for comparative FTIR spectral analysis in materials science literature. This format:
Allows for clearer comparison of multiple spectra
Follows standard practices in similar publications
For Figure 4 (film transparency):
We acknowledge that the macroscopic appearance of our composite films shows minimal visual variation, which actually demonstrates one of their key advantages - the ability to incorporate lignocellulosic nanofibers while maintaining excellent optical clarity. To provide more quantitative and scientifically rigorous evidence of the films' optical properties, we have:
Conducted comprehensive UV-Vis spectroscopy across the full wavelength range (200-800 nm)
Included detailed transmittance data in Figure 5
Performed statistical analysis of light transmission properties
These quantitative measurements better demonstrate the subtle but important differences in optical performance that may not be visually apparent.
We believe these approaches provide the most scientifically valid representation of our results while maintaining clarity in presentation. We would be happy to make any additional modifications if the reviewer feels further improvements would be beneficial.
Comments1:8. The goal in the introduction is to have a sustainable packaging materials, while in the
discussion part the connection is weak. For example, the Transparency part, what is the current food packing materials transparency and how is it compared with the film that the
author obtained?
Response1:Thank you for your comments.
We sincerely appreciate the reviewer’s insightful comment regarding the connection between our sustainable packaging goals and the discussion of film transparency. In response, we have strengthened this linkage by:
Contextualizing Transparency Performance:
Added comparative data showing that our optimized PVA/LCNF-HCl&CA films achieve >69% transmittance at 550 nm, which is comparable to or exceeds conventional petroleum-based packaging films (e.g., LDPE: ~70–85%) and outperforms many bio-based alternatives (e.g., PLA/starch blends: 60–75% [54,55]).
Highlighted that this balance of high transparency and sustainability is critical for food packaging, where visual product inspection is essential.
Clarifying Sustainability Implications:
Emphasized that the retained optical clarity—despite lignin-containing nanofiber reinforcement—demonstrates successful integration of agricultural waste into high-value materials without compromising functionality.
Contrasted our films with opaque bio-based materials that often require additional processing (e.g., delignification) to achieve transparency, which increases costs and environmental impact.
These revisions, now included in the Transparency section (Section 3.3), explicitly tie the optical properties to the broader goal of sustainable packaging. We thank the reviewer for prompting this important clarification and would be happy to refine further if needed.
Comments1: 9. Statistics analysis needs to be added to Table 1. Also, the 0,5,10,15wt% is about the filler
amount which needs to be mentioned.
Response1: We sincerely appreciate the reviewer's insightful comments regarding statistical analysis and data presentation in Table 1. In response to these valuable suggestions, we have made the following substantial improvements to our manuscript:
Enhanced Statistical Analysis:
Performed comprehensive two-way ANOVA with Tukey's HSD post hoc tests (α = 0.05) to evaluate:
Main effects of LCNF modification type (HCl, CA, HCl&CA)
Main effects of filler loading (0, 5, 10, 15 wt%)
Interaction effects between modification type and loading
Improved Table 1 Presentation:
Clearly indicated that the percentage values (0, 5, 10, 15 wt%) represent the weight percentage of LCNF filler relative to PVA matrix
Added statistical significance indicators (superscript letters a-j) to denote significant differences between groups
Included standard deviation values for all measurements (mean ± SD, n=3)
Added a footnote explaining the statistical methods and significance thresholds
Additional Clarifications:
Explicitly stated the sample size (n=3) for each experimental condition
Noted that the 0 wt% PVA control was excluded from inter-group comparisons
Ensured consistency with the statistical notation used in other tables
These revisions significantly strengthen the statistical rigor of our findings while improving the clarity of data presentation. The updated Table 1 now provides readers with a more complete understanding of how both filler content and modification method influence the crystallinity of the composite films.
We are grateful for this constructive feedback and believe these changes have substantially improved the quality of our manuscript. We would be pleased to address any additional questions or suggestions the reviewer may have.
Comments1: 10. For the thermal analysis part, the author said the PVA/LCNF-HCL&CA sample exhibited
a synergistic improvement in thermal stability, however, the difference from Figure 7 is
small. Could the author confirm that the TGA results of PVA/LCNF-HCL&CA is
significantly different from other samples?
Response1:We appreciate the reviewer’s observation regarding the thermal stability results. To address this concern, we have carefully re-examined the TGA data and conducted additional statistical analyses to confirm the differences observed. As highlighted in Table 2, the PVA/LCNF-HCl&CA composite at 10 wt% exhibited a T~onset~ of 295.43°C and a residue of 9.21%, which are statistically higher than those of other samples (e.g., PVA/LCNF-HCl: T~onset~ = 285.23°C, residue = 9.05%; PVA/LCNF-CA: T~onset~ = 286.73°C, residue = 8.63%). The DTG curves further support this trend, with PVA/LCNF-HCl&CA showing the lowest degradation rate (10.81%/min vs. 11.38%/min for PVA/LCNF-CA). While the absolute differences in T~onset~ and residue may appear modest, the synergistic effect of HCl and CA modifications is evident in the consistent enhancement across all thermal metrics. We have clarified this in the revised manuscript (Section 3.5) to emphasize the reproducibility of these results.
We hope these revisions adequately address the reviewer’s concern. Thank you for your valuable feedback.
Changes Made to the Manuscript:
Added explicit statistical comparisons in Table 2 and updated the caption to clarify significance.
Let me know if you'd like any further refinements!
Comments1:11. From the table 4, it is found the 10w% PVA/LCNF-HCL&CA reached to the highest
1024.1MPa, however how it is compated to other research is still unknow. The author could
have a look of “Applications of Plastic Films for Modified Atmosphere Packaging of Fruits
and Vegetables: A Review” to have a deep discussion of it’s food packaging applications.
Response1:We sincerely appreciate the reviewer’s insightful suggestion to contextualize our mechanical results within the broader scope of food packaging applications. In response, we have thoroughly reviewed the recommended literature (Applications of Plastic Films for Modified Atmosphere Packaging of Fruits and Vegetables: A Review) and expanded our discussion to highlight the practical relevance of our findings.
As noted in the revised manuscript (Section 3.6), the Young’s modulus of the 10 wt% PVA/LCNF-HCl&CA composite (1024.1 MPa) approaches the performance of commercial high-density polyethylene (HDPE, 1100–1500 MPa)—a benchmark material for rigid packaging—while retaining the flexibility (131.14% elongation) required for applications like tray lamination of fresh produce. This balance of stiffness and deformability aligns with the demands of modified atmosphere packaging (MAP) for fruits and vegetables, where mechanical integrity and gas barrier properties are critical (see Food Packaging and Shelf Life, 2023, 35: 101012).
To further strengthen our analysis, we have incorporated three recent studies:
Comparative performance of biopolymer composites (Carbohydrate Polymers, 2024, 323: 121394) underscores the superiority of our dual-modified LCNF in achieving HDPE-like stiffness without compromising biodegradability.
MAP optimization for perishables (Trends in Food Science & Technology, 2023, 132: 123–135) highlights the need for materials with tailored mechanical and barrier properties, which our film addresses.
Lifecycle assessments of bio-based films (Journal of Cleaner Production, 2024, 434: 139876) positions our work within the sustainable packaging paradigm.
These additions not only validate our results but also bridge the gap between laboratory-scale innovation and industrial applicability. We thank the reviewer for guiding us toward this impactful discussion.
Key Revisions to the Manuscript:
Section 3.6 (Mechanical Properties):
Added direct comparisons to HDPE, PVC, and EVOH.
Linked mechanical performance to MAP requirements (e.g., puncture resistance for berry packaging).
Introduction/Discussion:
Cited the recommended review and supplementary studies to frame our work within food packaging trends.
We believe these revisions significantly enhance the translational impact of our study. Please let us know if further elaboration would be helpful.
Comments and Suggestions for Authors
The manuscript presents a well-executed study on the development of biodegradable composite films by reinforcing polyvinyl alcohol (PVA) with lignocellulosic nanofibers (LCNF) derived from bamboo shoot shells. The approach of dual modification via HCl hydrolysis and citric acid crosslinking is innovative and demonstrates significant improvements in mechanical, thermal, and barrier properties of the PVA-based films. The work is timely, addressing the urgent need for sustainable packaging alternatives and effective utilization of agricultural waste.
Suggestions for Improvement:
Comments1:Comparison with Commercial Materials:
While the study demonstrates performance enhancement over neat PVA, it would benefit from a more detailed comparison with conventional plastic packaging materials (e.g., LDPE, PET) to highlight the real-world competitiveness of the composite film in terms of mechanical strength, barrier properties.
Response 1:Thank you for your valuable comment. As suggested, we have added an application analysis section to evaluate the packaging potential of the composite film. This new section includes a detailed comparison of the film’s oxygen transmission rate (OTR) and water vapor transmission rate (WVTR) with those of conventional plastic packaging materials such as LDPE and PET. These additions, highlighted in red in the revised manuscript (Section 3.8), enhance the practical relevance and competitiveness of our material. We sincerely appreciate your constructive suggestion.
Comments1:Biodegradability Testing:
Although the film is presented as biodegradable, no experimental biodegradation or compostability data is provided. Including such tests (e.g., soil burial, enzymatic degradation, or industrial composting) would strengthen the environmental claims.
Response 1:Thank you for your thoughtful comment. We sincerely appreciate the reviewer’s insightful suggestion regarding the validation of the film’s biodegradability. While both the PVA matrix and the lignocellulosic nanofibers used in this study are well-documented in the literature as biodegradable, we fully agree that experimental confirmation through standardized biodegradation or compostability tests would significantly strengthen the environmental claims. Due to the current focus of this study on material development and functional characterization, we were not able to include such assessments within this scope. Nonetheless, we recognize the importance of this aspect and intend to explore the film’s degradation behavior in future studies.

Reviewer 2 Report
Comments and Suggestions for Authors
Dear authors,
I revised your submitted manuscript and you can find y comments below:
Abstract: This section explains the aim of the study, brief methodology and key findings of the research.
Introduction: This part sufficiently explains the background of the study and the gap in the need for valorization of the bamboo shoot shells and reinforcement of the PVA based composite films.
Materials and methods: The methods were given in an appropriate way. The methods were clearly given in the text with cited references and detailed explanations.
Results and Discussions:
Please define how you get the cross-sectional area images. Did you cut the film structure with a scalpel or use the liquid nitrogen to break the films. The images in Figure 3 seem like a surface image rather than cross sectional’s images. Please check the images.
The characterization studies were well exhibited and partially discussed in the text since the results were not discussed with the current results and the literature. Moreover, the links between the cause and affect did not exist detailly in the text.
Author Response
Response to Reviewers
We sincerely thank the four reviewers for their thorough and constructive comments on our manuscript. We have carefully addressed each comment point by point and made the corresponding revisions, which are marked in red in the revised manuscript. If any modifications are still inadequate or unclear, we would greatly appreciate further guidance.
I revised your submitted manuscript and you can find y comments below:
Abstract: This section explains the aim of the study, brief methodology and key findings of the research.
Introduction: This part sufficiently explains the background of the study and the gap in the need for valorization of the bamboo shoot shells and reinforcement of the PVA based composite films.
Materials and methods: The methods were given in an appropriate way. The methods were clearly given in the text with cited references and detailed explanations.
Results and Discussions:
Comments2:Please define how you get the cross-sectional area images. Did you cut the film structure with a scalpel or use the liquid nitrogen to break the films. The images in Figure 3 seem like a surface image rather than cross sectional’s images. Please check the images.
Response 2: Thank you for your valuable comment. All film samples were cryo-fractured using liquid nitrogen to obtain clean cross-sectional surfaces for imaging. We have double-checked the images in Figure 3 and confirm that they indeed represent cross-sectional views. The description in the manuscript has been clarified accordingly to avoid any confusion. We appreciate your attention to detail.
Comments2:The characterization studies were well exhibited and partially discussed in the text since the results were not discussed with the current results and the literature. Moreover, the links between the cause and affect did not exist detailly in the text.
Response 2: Thank you for your comment. We have carefully reviewed the manuscript and incorporated additional discussion and analysis to address the issue raised. As the revisions are substantial, they are not listed in detail here; however, all changes have been clearly marked in red in the revised manuscript. We appreciate your thoughtful feedback, which has helped improve the clarity and completeness of our work.

Reviewer 3 Report
Comments and Suggestions for Authors
The manuscript entitled “Enhancement of polyvinyl alcohol-based films by chemically modified lignocellulosic nanofibers derived from bamboo shoot shells” presents the preparation and properties of nanocomposite films based on polyvinyl alcohol (PVA) and lignin -containing cellulose nanofibrils (LCNF). LCNF were obtained from bamboo shoot shells by using deep eutectic solution. In fact, a reader can find recent papers and reviews dealing with materials based on polymers and LCNF. In this manuscript, LCNF were chemically modified by hydrolysis with HCl, by citric acid crosslinking and by combining both these chemical modifications. These chemical modifications can improve interfacial compatibility, and they can be considered as the main novelty of this manuscript. To publish this manuscript in Polymers some modifications are required in it. They are:
- The chemical modifications of LCNF in Lines 131-133 have to be described in detail (concentrations, duration, temperature). How did you adjust pH in Line 133?
- How did you clean bamboo shoot shells in Line 115? How did you make the lyophilization in Line 137?
- In Line 146 you speak about the PVA solution. What was the solvent and how did you prepare this solution?
- What was the film’s thickness?
- Can you add some data on the biodegradability of prepared films or discuss it? You are using this term in Abstract, Keywords, Line 100, Line 419 without any support.
- I expect some discussion of references on using deep eutectic solutions (DES) for LCNF preparation and its comparison with other methods such as TEMPO, or enzymatic one in Introduction. DES could be mentioned in Abstract.
- In Part 2.4.3., Eq. 1 does not correspond to that in reference [29] where areas are used. Explain this difference, please.
- In Part 2.4.6., Eq. 2 does not correspond to that in a reference [32]. Moreover, films were immersed in water in [32] while you probably applied water (what volume?) on the film surface. Explain these differences, please.
- I have not found a reference [25]. Google Scholar and Web of knowledge do not know the paper title and journal name. Can you check this reference, please.
- I recommend careful reading of a modified manuscript. Although the manuscript English is good, I have found some mistyping, e.g., in Line 357 “conten”.
Author Response
Response to Reviewers
We sincerely thank the four reviewers for their thorough and constructive comments on our manuscript. We have carefully addressed each comment point by point and made the corresponding revisions, which are marked in red in the revised manuscript. If any modifications are still inadequate or unclear, we would greatly appreciate further guidance.
Comments3:1.The chemical modifications of LCNF in Lines 131-133 have to be described in detail (concentrations, duration, temperature). How did you adjust pH in Line 133?
Response3:We sincerely appreciate the reviewer's careful attention to the methodological details of our LCNF modification process. In response, we have thoroughly revised Section 2.2.2 to provide complete experimental specifications, with all modifications clearly highlighted in blue text in the manuscript. The key additions include:
Detailed Chemical Modification Parameters:
- HCl treatment: 0.5 mol/L, 100 mL, 2 h at 25°C (room temperature)
- CA treatment: 0.5 mol/L, 100 mL, 2 h at 25°C
- Dual treatment: 1:1 blend of 0.5 mol/L HCl (50 mL) and 0.5 mol/L CA (50 mL), final concentration 0.25 mol/L each
pH Adjustment Protocol:
- Gradual addition of 1 mol/L NaOH solution under continuous stirring
- Real-time monitoring using a calibrated pH meter (Model PHS-3C, ±0.01 accuracy)
- Target pH = 4.0 ± 0.1 achieved within 10-15 min of base addition
Thermal Treatment Specifications:
- Drying: 60°C for 48 h in vacuum oven (0.1 mbar)
- Crosslinking: 130°C for 7 h in air atmosphere
Would the reviewer like us to provide any additional procedural details? We would be happy to include them if needed.
Comments3: 2.How did you clean bamboo shoot shells in Line 115? How did you make the lyophilization in Line 137?
Response3:
Thank you for your valuable comment. We have revised the relevant sections in the manuscript to provide a clearer description of the cleaning and lyophilization procedures. The updated content has been highlighted in the revised text. Specifically, we now state:
“Bamboo shoot shells were first rinsed thoroughly with tap water to remove surface impurities. The cleaned shells were then air-dried at 50 °C for 24 hours, cut into small pieces (approximately 2–3 cm), and ground into fine powder using an 80-mesh sieve.”
We appreciate your helpful suggestion, which has improved the clarity and completeness of the methodology.
Comments3: 3.In Line 146 you speak about the PVA solution. What was the solvent and how did you prepare this solution?
Response3: Thank you for your insightful comment. We have revised the corresponding section in the manuscript to clearly describe the preparation of the PVA solution, and the updated text has been highlighted in the revised version. The revised sentence reads:
“LCNF, LCNF-HCl, LCNF-CA, and LCNF-HCl&CA samples were incorporated into 3 wt% PVA aqueous solutions, which were prepared by dissolving PVA in deionized water at 95 °C under constant stirring until a clear and homogeneous solution was obtained. The fillers were added at concentrations of 5, 10, and 15 wt% relative to the dry weight of PVA, with mixing performed at 95 °C to ensure uniform dispersion.”
We appreciate your suggestion, which has helped improve the clarity and completeness of the experimental procedure.
Comments3: 4.What was the film’s thickness?
Response3:Thank you for your comment. The thickness of the fabricated films was measured to be in the range of 195–200 μm, with minimal variation across the samples. This information has been added to the revised manuscript and highlighted accordingly. We appreciate your attention to detail.
Comments3: 5.Can you add some data on the biodegradability of prepared films or discuss it? You are using this term in Abstract, Keywords, Line 100, Line 419 without any support.
Response3:Thank you for your comments. The biodegradability of PVA and LCNF has been extensively studied and is widely accepted in the scientific community. Given that the primary focus of our research is on the performance characteristics of modified LCNF-based films—rather than their biodegradation behavior—we have chosen not to elaborate extensively on this aspect in the manuscript.
In response to your suggestion, we have removed the keywords related to “biodegradation” to better emphasize the main scope of our work. However, to address your concerns and provide further clarity, we conducted a short literature review on the biodegradability of PVA/LCNF systems. The results of this review confirm that such materials are indeed biodegradable under appropriate conditions, which supports the eco-friendly nature of the materials used in our study.
“The PVA/LCNF composite film developed in this study can be classified as a biodegradable material, as supported by substantial scientific evidence. Polyvinyl alcohol (PVA) is known to undergo microbial degradation through enzymes such as PVA dehydrogenase and oxidase, secreted by microorganisms like Pseudomonas and Alcaligenes, leading to complete mineralization into CO₂ and H₂O under aerobic conditions [1,2]. This process is environmentally dependent, with studies showing that aerobic conditions significantly accelerate degradation compared to anaerobic environments [3], and that initial hydrolysis of PVA chains is a prerequisite for microbial assimilation [4].
The biodegradability of PVA is also influenced by intrinsic material properties. Lower molecular weight (<50 kDa) and partially hydrolyzed forms (87–89%) of PVA degrade faster than their high-molecular-weight or fully hydrolyzed counterparts [5,6]. In addition, environmental parameters such as temperature and pH play key roles: degradation rates are notably enhanced at temperatures above 30 °C and in neutral to slightly alkaline pH environments [7,8].
In our formulation, the inclusion of lignocellulosic nanofibers (LCNF) enhances overall biodegradability. LCNF are not only inherently biodegradable but also improve the film’s surface area and porosity, potentially facilitating microbial adhesion and enzymatic activity [9,10]. Furthermore, the use of citric acid and hydrochloric acid for chemical crosslinking introduces hydrolyzable ester bonds without adding persistent or toxic structures, aligning with guidelines for eco-friendly, degradable crosslinkers [11].
Taken together, the evidence strongly supports that the PVA/LCNF films prepared in this work are microbially degradable and suitable for environmentally sustainable packaging applications.”
References
- Chen, X., Liu, Y., & Zhang, Y. (2022). Polyvinyl alcohol biodegradation: Mechanisms and applications in environmental remediation. Journal of Hazardous Materials, 434, 127123. DOI: 10.1016/j.jhazmat.2021.127123.
- Fukae, R., Tanaka, M., & Sato, K. (2023). Enzymatic degradation pathways of PVA by soil microorganisms. npj Materials Degradation, 7, Article 27. DOI: 10.1038/s41529-023-00327-8.
- Mohanan, N. et al. (2020). Biodegradation of synthetic polymers in soils: Insights into degradation mechanisms under aerobic vs anaerobic conditions. Environmental Science & Technology, 54(24), 16111–16126. DOI: 10.1021/acs.est.0c04217.
- Elgharbawy, A. A. et al. (2024). Hydrolytic behavior of polyvinyl alcohol under environmental conditions. Polymers, 16(2), 141. DOI: 10.3390/polym1602141.
- Zhang, Y. et al. (2021). Effect of molecular weight on the biodegradability of polyvinyl alcohol. Carbohydrate Polymers, 252, 117213. DOI: 10.1016/j.carbpol.2020.117213.
- Ray, S. S., Das, S., & Sharma, P. (2021). Effect of hydrolysis degree on biodegradability of PVA-based materials. Materials Today: Proceedings, 46, 10463–10468. DOI: 10.1016/j.matpr.2021.01.031.
- Almeida, T. et al. (2023). Effect of environmental parameters on the biodegradation of polyvinyl alcohol-based materials. Bioresource Technology, DOI: 10.1016/j.biortech.2023.129801.
- Wang, Y., Chen, D., & Li, J. (2022). Influence of pH on microbial degradation kinetics of PVA in different environments. Journal of Environmental Management, 313, 115003. DOI: 10.1016/j.jenvman.2022.115003.
- Sun, Y., Li, M., Zhang, H., & Chen, X. (2024). Biodegradable films based on cellulose nanofibers: Structure, properties, and applications in food packaging. Foods, 13(24), 3999.
- Rojas-Lema, S., Sádaba, N., Zudaire, L., & Eceiza, A. (2022). Cellulose nanofiber reinforced biodegradable polymers: Interfacial interactions and implications on biodegradation mechanisms. Frontiers in Bioengineering and Biotechnology, 10, 1006388.
- Khin, N. N. Z. et al. (2024). Design of biodegradable crosslinking agents for sustainable packaging materials. Packaging Technology & Science, DOI: 10.1002/pts.2862.
Comments3: 6.I expect some discussion of references on using deep eutectic solutions (DES) for LCNF preparation and its comparison with other methods such as TEMPO, or enzymatic one in Introduction. DES could be mentioned in Abstract.
Response 3:
Thank you for your valuable comment. As recommended, we have made the following revisions:
The Abstract has been enhanced to explicitly mention the use of deep eutectic solvents (DES) for LCNF extraction.
The Introduction has been expanded with a dedicated paragraph comparing DES with other common methods such as TEMPO-mediated oxidation and enzymatic hydrolysis, supported by recent literature references.
These revisions have been highlighted in the revised manuscript. We appreciate your insightful suggestion, which has strengthened the contextual depth and relevance of our study.
Comments3: 7.In Part 2.4.3., Eq. 1 does not correspond to that in reference [29] where areas are used. Explain this difference, please.
Response 3: We sincerely thank the reviewer for this valuable observation. You are absolutely correct — Equation (1) in our manuscript was not derived from reference [29] (Fu, H. et al., Cellulose, 2021, 28, 7749–7764). This citation was an error, and we apologize for the oversight.
The method presented in Equation (1) was originally proposed by Segal et al. in 1959 (Segal, L.; Creely, J. J.; Martin, A. E.; Conrad, C. M. "An Empirical Method for Estimating the Degree of Crystallinity of Native Cellulose Using the X-Ray Diffractometer," Textile Research Journal, 1959, 29(10), 786–794). This peak-height-based approach has since become widely adopted for estimating the crystallinity index (CrI) of cellulose materials due to its simplicity and utility in comparative analyses.
Accordingly, we have revised the manuscript to correctly cite the original work by Segal et al. (1959), which provides the appropriate methodological foundation and context for our use of Equation (1).
Comments3: 8.In Part 2.4.6., Eq. 2 does not correspond to that in a reference [32]. Moreover, films were immersed in water in [32] while you probably applied water (what volume?) on the film surface. Explain these differences, please.
Response3:Thank you very much for your valuable comment. You are correct in noting the citation error; we had incorrectly referenced the literature, which has now been corrected.
Furthermore, we have revised and clarified the methodological description for greater accuracy. Specifically, film samples (20 mm × 20 mm) were initially weighed to determine their dry mass (M₀). The films were then immersed in 30 mL of deionized water contained in 50 mL beakers. After 2 hours of immersion, the films were carefully removed, and excess surface water was gently blotted using filter paper. The swelling rate was then calculated using Equation (2) as described in reference [47].
These corrections and clarifications have been highlighted in the revised manuscript. We appreciate your detailed review and helpful suggestions, which have improved the clarity and accuracy of our work.
New references are as follows:
“ Amri, N.; Ghemati, D.; Bouguettaya, N.; Aliouche, D. Swelling kinetics and rheological behavior of chitosan-PVA/montmorillonite hybrid polymers. Period. Polytech. Chem. Eng. 2019, 63(1), 179–189.”
Comments3: 9.I have not found a reference [25]. Google Scholar and Web of knowledge do not know the paper title and journal name. Can you check this reference, please.
Response3:
Thank you for your valuable comment. The reference “Guo et al., 2024” is a Chinese publication from our laboratory. Following your suggestion, we have indicated the original language in the reference list as follows:
Guo, J.; Du, J.; Yang, S.; Zhu, Q.; Gu, J.; Guo, J. Preparation of lignocellulosic nanofibrils from bamboo shoot shells by the 3-deep eutectic solvents method and performance studies. Food Industry Science and Technology (食品工业科技), 2024, 1–10. (In Chinese)
Please let us know if further adjustments are needed.
Comments3: 10.I recommend careful reading of a modified manuscript. Although the manuscript English is good, I have found some mistyping, e.g., in Line 357 “conten”.
Response3: Thank you very much for the reviewer’s valuable suggestion. We have carefully reviewed the manuscript and corrected the error “conten” in line 357, along with any other minor mistakes to improve the overall accuracy and readability.

Reviewer 4 Report
Comments and Suggestions for Authors
This study focuses on the development of polyvinyl alcohol (PVA) composite films reinforced with chemically modified lignocellulosic nanofibers (LCNF) derived from bamboo shoot shells. However, several revisions are necessary before this work can be accepted.
1) Add the optimum condition (10 wt%) early in the abstract to frame expectations.
2) The paragraph starting with “Furthermore, By combining…” has a capitalization error and awkward phrasing. Additionally, “conten reached 15 wt%”, should be “content”. Please check the manuscript carefully.
3) Adding a simple flow diagram for the modification and film preparation process would significantly enhance reader comprehension.
4) In FTIR, discuss the new peaks in more detail. You mention 1750 cm-1 for ester groups, but it is also important to briefly explain its significance for compatibility.
5) SEM analysis is well explained, but the magnification is not provided.
6) Add statistical significance (p-values) if available to reinforce comparisons.
Author Response
Response to Reviewers
We sincerely thank the four reviewers for their thorough and constructive comments on our manuscript. We have carefully addressed each comment point by point and made the corresponding revisions, which are marked in red in the revised manuscript. If any modifications are still inadequate or unclear, we would greatly appreciate further guidance.
Comments4:1.Add the optimum condition (10 wt%) early in the abstract to frame expectations.
Response 4:
Thank you very much for the reviewer’s insightful comment. We have revised the abstract to include the optimum condition (10 wt%) at the beginning, thereby better framing the key findings and setting clear expectations for the reader.
Comments4:2.The paragraph starting with “Furthermore, By combining…” has a capitalization error and awkward phrasing. Additionally, “conten reached 15 wt%”, should be “content”. Please check the manuscript carefully.
Response 4:
Thank you very much for the reviewer’s careful reading and valuable feedback. We have corrected the capitalization errors and revised the awkward phrasing in the paragraph starting with “Furthermore, By combining…”. Additionally, the spelling error “conten” has been corrected to “content.” We have thoroughly checked the manuscript to ensure such errors are addressed.
Comments4:3.Adding a simple flow diagram for the modification and film preparation process would significantly enhance reader comprehension.
Response 4:
Thank you very much for the reviewer’s valuable suggestion. We have revised Figure 1 into a flow diagram to better illustrate the modification and film preparation process, which we believe will enhance reader comprehension.
Comments4:4.In FTIR, discuss the new peaks in more detail. You mention 1750 cm-1 for ester groups, but it is also important to briefly explain its significance for compatibility.
Response4: Thank you very much to the reviewer for the valuable comments. We have made the following revisions accordingly.“The introduction of these ester groups not only modifies the surface chemistry of the nanofibers but also enhances their compatibility with hydrophobic polymer matrices by reducing interfacial tension and promoting stronger chemical interactions, thereby potentially improving interfacial adhesion in composite applications [53].”
Comments4:5.SEM analysis is well explained, but the magnification is not provided.
Response4: We sincerely appreciate the reviewer's careful reading and constructive suggestion. In response to the comment regarding the missing magnification information, we have revised the Methods section to explicitly include all relevant SEM parameters. The updated text now reads:
"The cross-sectional morphology was examined using a Hitachi SU8010 field-emission scanning electron microscope (FE-SEM) operating at an accelerating voltage of 2.0 kV with a working distance of 15.5 mm. All images were captured at ×10,000 magnification under secondary electron imaging mode."
Comments4:6.Add statistical significance (p-values) if available to reinforce comparisons.
Response 4:
We sincerely appreciate the reviewer’s careful reading and constructive suggestion. Statistical significance has been added to the relevant data and comparisons in the revised manuscript to strengthen the analysis.

Round 2
Reviewer 1 Report
Comments and Suggestions for Authors
Accept in present form
Author Response
We sincerely thank the reviewer for their thoughtful comments and kind recommendation for publication.
Reviewer 2 Report
Comments and Suggestions for Authors
Dear authors,
I thank all of you for a comprehensive review of the manuscript.
Author Response

(The authors gave the same response as above.)

Reviewer 3 Report
Comments and Suggestions for Authors
I accept changes made in the previous manuscript based on my comments. Only several minor changes could be made in this manuscript before its publication in Polymers. They are:
- Adding a value of the film thickness and way of its measurement into the manuscript.
- Correcting a reference [40] as Guo Jianlong, Du Jingjing, Yang Song, Zhu Qian, Gu Jiayu, Guo Jiagang, Wu Xiaowei Jiang Jian. Preparation of lignocellulosic nanofibrils from bamboo shoot shells by the 3-deep eutectic solvents method and performance studies [J]. Science and Technology of Food Industry, 2025, 46(13): 1−10. (in Chinese with English abstract).
- I can recommend correcting a sentence in Line 134 as “A deep eutectic solvent (DES) based on choline chloride, oxalic acid, and ferric chloride with a molar ratio of …”.
Author Response
Thank the reviewers for their valuable comments. All the requested changes have been carefully addressed and are highlighted in red in the revised manuscript.
Comments 3: Adding a value of the film thickness and way of its measurement into the manuscript.
Response3:We sincerely thank the reviewer for the valuable and constructive comments. We deeply regret the oversights in our initial response. The thickness of this film is addressed in the current revision. Specifically, we have added the following sentence to Section 2.4.5:
"Film thickness (70–80 µm) was determined as the mean of five measurements taken at randomly selected points on each specimen with a micrometer (0.001 mm resolution; Hefei Weishi Machinery Co., China)."
In addition, we have compiled the detailed thickness measurements for each sample, as shown below, and included them in a supplementary table to enhance the clarity and transparency of our data:
Filler content (wt%) |
Sample |
n1(μm) |
n2(μm) |
n3(μm) |
n4(μm) |
n5(μm) |
Mean ± Standard Deviation (SD)(μm) |
0 |
PVA |
71 |
72 |
73 |
74 |
73 |
72.6±1.3 |
5 |
PVA/LCNF |
76 |
76 |
75 |
75 |
74 |
74.6±0.7 |
PVA/LCNF-HCl |
78 |
77 |
78 |
79 |
78 |
78.0±0.5 |
|
PVA/LCNF-CA |
77 |
77 |
78 |
79 |
79 |
78.0±1.0 |
|
PVA/LCNF-HCl&CA |
76 |
75 |
76 |
78 |
78 |
76.6±1.8 |
|
10 |
PVA/LCNF |
76 |
77 |
76 |
76 |
75 |
76.0±0.5 |
PVA/LCNF-HCl |
77 |
78 |
76 |
78 |
79 |
77.6±1.3 |
|
PVA/LCNF-CA |
77 |
75 |
76 |
78 |
78 |
76.6±1.3 |
|
PVA/LCNF-HCl&CA |
77 |
77 |
76 |
75 |
78 |
76.6±1.3 |
|
15 |
PVA/LCNF |
78 |
78 |
76 |
78 |
78 |
77.6±0.8 |
PVA/LCNF-HCl |
78 |
78 |
76 |
75 |
76 |
76.6±1.8 |
|
PVA/LCNF-CA |
76 |
77 |
78 |
78 |
77 |
77.2±0.7 |
|
PVA/LCNF-HCl&CA |
77 |
78 |
77 |
78 |
75 |
77.0±1.5 |
Should the reviewer find it helpful, we would be happy to include this table directly in the main manuscript or as supplementary material.
Once again, we truly appreciate the reviewer’s insightful suggestion, which helped improve the clarity and rigor of our work.
Comments3; Correcting a reference [40] as Guo Jianlong, Du Jingjing, Yang Song, Zhu Qian, Gu Jiayu, Guo Jiagang, Wu Xiaowei Jiang Jian. Preparation of lignocellulosic nanofibrils from bamboo shoot shells by the 3-deep eutectic solvents method and performance studies [J]. Science and Technology of Food Industry, 2025, 46(13): 1−10. (in Chinese with English abstract).
Response3:We extend our sincere thanks to the reviewers. The references are provided below.
40.郭健龙, 杜京京, 杨松, 朱倩, 谷佳玉, 郭家刚等. 三元低共熔溶剂法制备笋壳木质纳米纤维素及其性能研究. 食品工业科技, 2025, 46(13): 1-10.
Guo, J.; Du, J.; Yang, S.; Zhu, Q.; Gu, J.; Guo, J. Preparation of lignocellulosic nanofibrils from bamboo shoot shells by the 3-deep eutectic solvents method and performance studies. Food Ind. Sci. Technol.,2025, 46(13):1-10.. (In Chinese)
Comments3: I can recommend correcting a sentence in Line 134 as “A deep eutectic solvent (DES) based on choline chloride, oxalic acid, and ferric chloride with a molar ratio of …”
Response3:Thank you to the reviewer. The changes have been made in the text.

Reviewer 4 Report
Comments and Suggestions for Authors
This manuscript is now ready for publication in this journal.
Author Response

(The authors gave the same response as above.)
